# Thermal Infrared Small Ship Detection in Sea Clutter Based on Morphological Reconstruction and Multi-Feature Analysis

**Yongsong Li** [1,2], **Zhengzhou Li** [1,2,3,*], **Yong Zhu** [1,2], **Bo Li** [1,2], **Weiqi Xiong** [1,2] **and Yangfan Huang** [1,*]

1   School of Microelectronics and Communication Engineering, Chongqing University, Chongqing 400044, China
2   Key Laboratory of Dependable Service Computing in Cyber Physical Society of Ministry of Education, Chongqing University, Chongqing 400044, China
3   Key Laboratory of Beam Control, Institute of Optics and Electronics, Chinese Academy of Sciences, Chengdu 610209, China
*   Correspondence: lizhengzhou@cqu.edu.cn (Z.L.); hyf@cqu.edu.cn (Y.H.); Tel.: +86-132-0601-5717 (Z.L.)

**Abstract:** The existing thermal infrared (TIR) ship detection methods may suffer serious performance degradation in the situation of heavy sea clutter. To cope with this problem, a novel ship detection method based on morphological reconstruction and multi-feature analysis is proposed in this paper. Firstly, the TIR image is processed by opening- or closing-based gray-level morphological reconstruction (GMR) to smooth intricate background clutter while maintaining the intensity, shape, and contour features of ship target. Then, considering the intensity and contrast features, the fused saliency detection strategy including intensity foreground saliency map (IFSM) and brightness contrast saliency map (BCSM) is presented to highlight potential ship targets and suppress sea clutter. After that, an effective contour descriptor namely average eigenvalue measure of structure tensor (STAEM) is designed to characterize candidate ship targets, and the statistical shape knowledge is introduced to identify true ship targets from residual non-ship targets. Finally, the dual method is adopted to simultaneously detect both bright and dark ship targets in TIR image. Extensive experiments show that the proposed method outperforms the compared state-of-the-art methods, especially for infrared images with intricate sea clutter. Moreover, the proposed method can work stably for ship target with unknown brightness, variable quantities, sizes, and shapes.

**Keywords:** thermal infrared (TIR) imaging; small ship target detection; sea clutter; gray-level morphological reconstruction; saliency detection; multi-feature analysis

## 1. Introduction

Infrared ship target detection is an important technology for maritime search and track applications [1,2], where both accuracy and robustness are indispensable. However, because of the long imaging distance, for thermal infrared (TIR) images, the signal intensity of a small ship target is usually very weak without sufficient texture and structure information. More importantly, the complicated sea clutter such as sun glint, tail wave, island, sea fog and sea-sky line are usually capricious without predictable shape, which reduces the accuracy of TIR ship target detection. Moreover, the variable size and irregular shape of a ship target also further restrict the robustness of target detection. For above-mentioned reasons, infrared ship detection has attracted many researchers and a number of ship target detection algorithms are designed [3–5].

The representative TIR ship detection algorithms can be roughly divided into two categories, namely, detect-before-track (DBT) strategy and track-before-detect (TBD) strategy. By taking full advantage of the continuity of moving target and the stationarity of background, the TBD-based detection methods, such as frame difference method [6], three-plot correlation filter [7] and multi-stage hypothesis test filter [8], have achieved outstanding performance for the ships with continuous trajectory. However, in many maritime scenes, the background is changeable, and the target trajectory might be discontinuous, so the performance of TBD methods would degrade sharply. Compared with TBD strategy-based methods, the DBT-based methods have many advantages, such as less prior knowledge and faster computational speed and can work stably for the target without continuous trajectory under variable background. Therefore, the DBT-based ship detection methods are of great significance and have been drawing much attention from researchers recently. The existing DBT-based ship detection methods can be approximately classified into four categories: The target/background modeling-based methods, the image segmentation-based methods, the human visual system (HVS)-based methods, and the machine learning-based methods.

In the target/background modeling-based methods, the ship target and background are separated by modeling the TIR imaging properties of ship target or background. Lagaras et al. [9] introduced an end-to-end temperature difference model for the detection and classification of TIR ships at the environmental conditions with an analysis-based scanning detector. Wang et al. [10] proposed a TIR ship detection method by modeling ship radiation anomalies with a nonlinear statistical Gaussian mixture model. These methods try to extract the ship target from background via modeling on the premise of acquiring abundant knowledge about the infrared radiation characteristics of ship target and background, but it is difficult to meet these requirements. To solve this problem, Kim et al. [11,12] estimated the background and distinguished the target from sea clutter by using the heterogeneous background removal filter. Furthermore, the statistical histogram curve transforms were also developed for the infrared maritime target detection based on the model assumption that the ship target region is much brighter than the background, such as the methods mentioned in [13,14]. These background removal filters and statistical histogram curve transforms are excellent for the infrared point ship target with relatively high positive contrast. However, for low contrast or negative contrast, the ship targets are dim, and the background clutter is intricate, so these methods may rapidly reduce the detection performance.

The image segmentation-based methods have the advantages of simplicity and efficiency, which are widely used for ship detection in TIR images. The classical threshold segmentation methods such as 2D Otsu [15], minimum error [16], and 2D maximum entropy [17] are well known in infrared ship target segmentation for their simplicity and easy-implementation. Nevertheless, these classical threshold segmentation methods are sensitive to noise and cannot detect small or low contrast ship targets due to the fact that their performance is easily affected by the clutter intensity information. To overcome the disturbance of noise and background clutter, the mean shift-based ship segmentation algorithms developed in [18,19] have achieved considerable detection performance for infrared ship in sea clutter. Whereas, because the mean shift methods are based on region clustering and merging, they may obtain a wrong detection when the ship target has low contrast or point size. In addition, the active contour-based Chan–Vese models [20,21] are also commonly used in the field of ship target detection because they can effectively segment the targets in homogenous background by extracting topology structure. Unfortunately, for complex background with heterogeneous sea clutter, the low-contrast ship target and sea clutter might be similar in topology structure, so serious false detection might happen.

The human visual system-based methods are based on the local contrast measure and selective attention mechanism of the ship target region, and therefore, the feature saliency map calculation of infrared ship target is the foremost topic for HVS-based methods. Mumtaz et al. [22] adopted graph-based visual saliency (GBVS) method to compute a saliency map, and then extracted the ship target by using multilevel thresholding of the saliency map. The GBVS-based method may extract strongly salient clutter regions and fail to detect real ship target when the sea clutter is heavy.

To conquer this problem, Liu et al. [23,24] proposed an effective infrared ship target detection method based on saliency map fusion by exploiting multi-features of ship target, including local contrast, edge information, and brighter intensity. Following, Bai et al. [25] presented a new detection method for low-contrast infrared ship targets by analyzing the fuzzy inference system that integrates both local saliency information and global spatial feature. The two methods can acquire excellent performance for the detection of infrared ships in complex background clutter. However, these methods are based on the assumption that target regions are comparatively brighter than the dark sea surface, so they cannot detect the negative-source dark ship targets submerged in relatively bright backgrounds.

In the machine learning-based methods, the infrared ship detection problem is considered as a two-class (ship target and background) recognition problem. In these methods, the infrared images are depicted by multiple feature vectors, and then the ship target class and background class can be distinguished by classifiers, such as feedforward neural network [26], extreme learning machine [27], artificial neural network [28], and convolutional neural network [29,30]. These machine learning-based methods can easily detect diverse ship targets under intricate background clutter in some cases. However, regrettably, these methods must spend plenty of time training samples and selecting features. Moreover, in practical TIR maritime applications, these machine learning-based methods fail to generate enough training samples due to the complexity and variability of the sea clutter, leading to the deterioration of the ship target detection capability.

Comparing the advantages and disadvantages of above-mentioned methods, although many studies have been focused on the detection of TIR ship targets against complex backgrounds in the past decades, it is still an open issue. Actually, in TIR remote sensing images, the natural scene background has strong local self-similarity, but the small ship target as a manmade object will destroy these characteristics of background, so compared with the background clutter even for heavy sea clutter, the ship target has solid intensity, contrast, contour, and shape features. To further overcome the disturbance of heavy sea clutter on the detection of ship targets with low-contrast, multiple targets, unknown brightness, and different sizes, we propose an effective and robust small ship detection scheme, based on the morphological reconstruction and multi-feature analysis of TIR imaging characteristics between ship targets and background clutter. Firstly, a pre-processing procedure based on closing (opening)-based gray-level morphological reconstruction (GMR) is introduced to remove noise and smooth intricate background clutter while preserving the ship target signals including intensity, shape, and contour information. Secondly, according to the intensity and local region contrast features of TIR ship targets, the intensity foreground saliency map (IFSM) and brightness contrast saliency map (BCSM) are computed and fused to well enhance potential ship targets, and an adaptive threshold is applied to segment candidate ship targets. Then, motivated by the contour characteristic of target region in GMR pre-processed image, a novel contour descriptor of TIR ship target named as average eigenvalue measure of structure tensor (STAEM) is proposed to characterize candidate ship targets and eliminate residual clutter simultaneously. After that, based on the statistics and observation of shape parameters of TIR ship targets selected from a comprehensive ship database namely visible and infrared ships (VAIS) [31], shape knowledge is obtained and utilized to further distinguish true ship targets from non-ship targets. Finally, the dual approach is adopted to detect both bright and dark ship targets in TIR image simultaneously. Extensive experiments show that the proposed ship target detection scheme outperforms the compared state-of-the-art algorithms under diverse backgrounds, and is suitable for ship targets with unknown brightness, variable sizes, and quantities.

Figure 1 gives the flow chart of the proposed TIR small ship target detection method. The red boxes have enlarged true ship targets, and the purple boxes have marked several highest STAEM values of false targets. The STAEM value of ship target is 0.3281, and the four largest STAEM values of false targets are 0.1431, 0.1149, 0.0960, and 0.0792, respectively. $\otimes$ denotes pixel-wise multiplication manner, and $\oplus$ denotes pixel-wise addition manner. Figure 1a is the input TIR ship image, and the final automatically detected ship target image is shown in Figure 1g. During this process, a dual approach for both bright and dark ship target detection is introduced. Figure 1b1–b4 shows the

GMR-based pre-processed images, and Figure 1c1–c4 gives the saliency detection results of IFSM and BCSM. Figure 1(d1,d2) shows the final fused saliency maps of bright and dark ship target, respectively. Figure 1(e1,e2) is the step of extracting candidate ship targets and eliminating residual clutter by STAEM. Figure 1(f1,f2) shows the detected bright and dark ship target maps after two-step ship verification method, respectively.

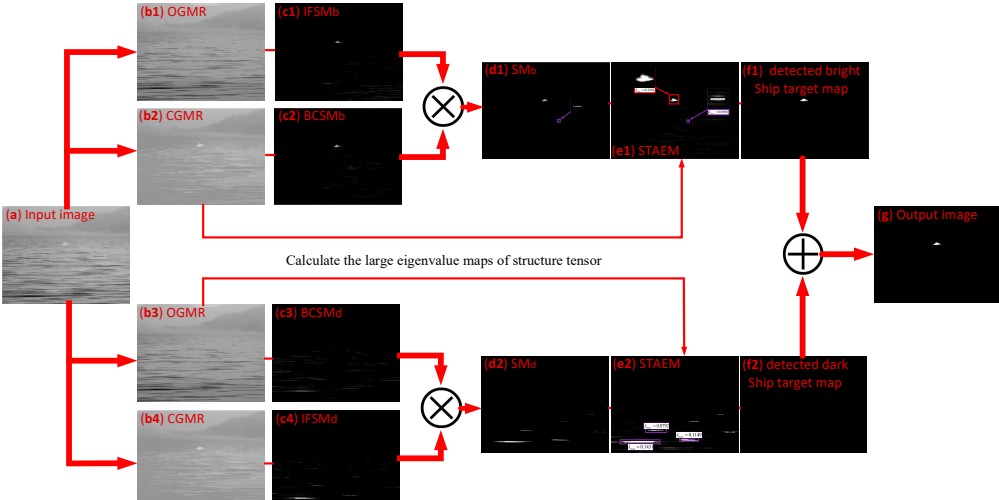

**Figure 1.** The flow chart of the proposed thermal infrared (TIR) small ship target detection method.

There are four contributions in this paper: (1) Traditional infrared ship target detection methods suffer the disturbance of heavy sea clutter. In this paper, the GMR-based pre-processing procedure is introduced to efficiently remove noise and smooth intricate background clutter. Moreover, to deeply explore the intensity and local region contrast features of TIR ship targets after GMR-based pretreatment, the IFSM and BCSM are computed and fused to highlight potential ship targets and suppress sea clutter. As far as we know, it is the first time that gray-level morphological reconstruction is used for suppressing heavy sea clutter and perceiving the saliency map detection for potential ship targets. (2) The STAEM is presented as a valid contour descriptor to further depict the candidate ship targets and eliminate residual clutter simultaneously. The proposed STAEM is a novel measure to describe the contour information of a connected region. (3) Based on the statistics and observation of the shape parameters of TIR ship targets selected from VAIS database, a statistical shape knowledge is generated and utilized to further extract true ship targets from candidate targets. Because the VAIS database contains 1242 TIR ship images composed of 264 different types of ships captured during the daytime and nighttime with variable view-angles and diverse distances, the shape knowledge obtained by statistics and observation on this database could be more widely used in the field of TIR small ship detection. (4) Combining the above methods and their advantages, an efficient and robust infrared small ship detection scheme is developed and is superior to the state-of-the-art ship target detection methods.

The structure of this paper is organized as follows: The morphological reconstruction and multi-feature analysis based on the intrinsic TIR imaging characteristics between ship targets and sea clutter are discussed in Section 2. The TIR ship detection algorithm based on morphological reconstruction and multi-feature analysis is proposed, and the whole details of the novel and robust scheme are shown in Section 3. Extensive experiments are included in Section 4 to evaluate the performance of the proposed algorithm, and the results show that the ship detection performance by the proposed method is significantly enhanced. Finally, Section 5 gives the conclusions of this paper.

## 2. Morphological Reconstruction and Multi-feature Analysis for TIR Ship Images

### 2.1. Characters of Small Ship Targets and Sea Clutter in TIR Image

TIR images and visible (VIS) or near-infrared (NIR) images are both numerical representations of electromagnetic radiation in a specific wavelength region, so they have many common grounds. The VIS spectra are the visible part of the electromagnetic spectrum with a wavelength of 0.38~0.75 μm. The infrared spectra can be roughly divided into: Near-infrared (NIR, wavelength 0.75~3 μm) and thermal infrared (TIR, 3~15 μm). Radiation emitted by an object spans a series of wavelengths, but because a specific sensor usually only collects radiation within a specific bandwidth, it is only interested in a limited range of the spectrum. According to Wien's displacement law [32,33], the peak emission wavelength $\lambda_{\max}$ of the radiation is inversely proportional to the absolute temperature $T$ of the object, and can be simply written as:

$$\lambda_{\max} T = 2898.8 \ \mu\text{m K} \tag{1}$$

According to this law, we can compute the temperature of blackbody radiation at each wavelength, and the temperatures of blackbody of VIS, NIR, and TIR are listed in Table 1. As can be seen from Table 1, the temperatures of blackbody of VIS and NIR spectra are very high, so they have little or no thermal imaging characteristics in natural conditions. In fact, the VIS is visible to the human eye with color perception. NIR is different from thermal imaging, and it is more like the extension of the grayscale of VIS imaging, so NIR is sometimes called "reflected infrared". Since the temperature range of blackbody of TIR is −79.9~693.12 °C, it is often used as thermography in natural conditions. Therefore, TIR imaging also has different physical characteristics from VIS light and NIR imaging.

**Table 1.** The temperatures of blackbody of visible (VIS), near-infrared (NIR), and TIR.

| Radiations | Wavelength (μm) | Temperatures of Blackbody (K) | Temperatures of Blackbody (°C) |
|:---:|:---:|:---:|:---:|
| VIS | 0.38~0.75 μm | 3865.07~7628.42 | 3591.92~7355.27 |
| NIR | 0.75~3 μm | 966.27~3865.07 | 693.12~3591.92 |
| TIR | 3~15 μm | 193.25~966.27 | −79.9~693.12 |

There are several special characteristics of small ship targets and sea clutter in TIR images. Firstly, the pixel intensity value in TIR image has a different physical significance with the pixel value in VIS or NIR images. The pixel intensity represents the thermal intensity ("heat") of objects. In TIR images, because different materials have different thermal radiation, the hotter object appears "white" while the cooler object appears "black". Due to this characteristic, the small ship targets as a manmade object in sea clutter can naturally form bright or dark objects in the TIR image despite the ship camouflage colors and the illumination condition in VIS or NIR images. Secondly, owing to the limited manufacturing level of sensors, the TIR images are usually less informative than the images captured by VIS or NIR optical sensors. In other words, the TIR images are relatively lower in resolution, contrast, and clarity, resulting in the lack of textural and structural information about objects. Thirdly, because thermal infrared radiation determines the temperature of objects, the infrared thermography can be used to remotely detect the ship targets. To conclude, because of the long imaging distance for TIR images, the signal intensity of a small ship target is usually very weak without sufficient texture and structure information. Furthermore, in TIR images, the natural scene background has strong local self-similarity even for sea clutter, but the small ship target as a manmade object will destroy these characteristics of background. Therefore, the ship targets can be viewed as local bright or dark uniform abnormal regions under the sea background clutter in TIR images due to long imaging distance, as illustrated in Figure 2.

TIR small ship imaging results are complex and comprehensive processes, which are easily affected by non-stationary inputs, such as atmospheric radiation, solar refraction and engine temperatures. Therefore, detection is very important when the ship target is small and embedded in sea clutter.

Figure 2 shows three representative TIR ship images with complex sea background clutter, in which it is difficult to detect all small ship targets because of low signal-to-clutter ratio (SCR), fewer pixels, heavy sea clutter, and lack of textural and structural information. The small ship targets are enlarged as corresponding three-dimensional (3-D) mesh plots by red boxes. Nevertheless, according to above character analysis, as the ship targets can be viewed as local bright or dark uniform abnormal regions under the sea clutter in TIR images due to long imaging distance, the small ship targets still have some solid features. Firstly, because the small ship targets (as a manmade object) and sea clutter have different thermal radiation characteristics, the ship targets have a certain pixel intensity feature against background sea clutter. Secondly, as depicted by the enlarged 3-D mesh plots in three TIR images, as small ship target will destroy the local self-similarity characteristics of sea background clutter, the ship targets have obvious local contrast and contour features compared with sea clutter. Thirdly, as shown in Figure 2a,b, as the small ship targets are almost uniform regions in sea clutter, they have obvious shape features. Therefore, based on these TIR imaging features of small ship targets and sea clutter, we propose an effective TIR small ship target detection scheme by reasonably integrating these features. As a primary step, a pre-processing procedure based on gray-level morphological reconstruction is introduced to remove noise and smooth sea clutter. Next, according to the intensity and local contrast features of ship targets, the IFSM and BCSM are presented and fused to well enhance potential ship targets and suppress heavy sea clutter. Then, motivated by the contour feature of target region, the STAEM-based contour descriptor is proposed to characterize candidate ship targets and eliminate residual clutter simultaneously. After that, based on the statistics and observation of shape parameters of TIR ship targets selected from VAIS dataset, the statistical shape knowledge is obtained and utilized to further distinguish true ship targets from non-ship targets. Finally, the whole proposed small ship target detection scheme is presented.

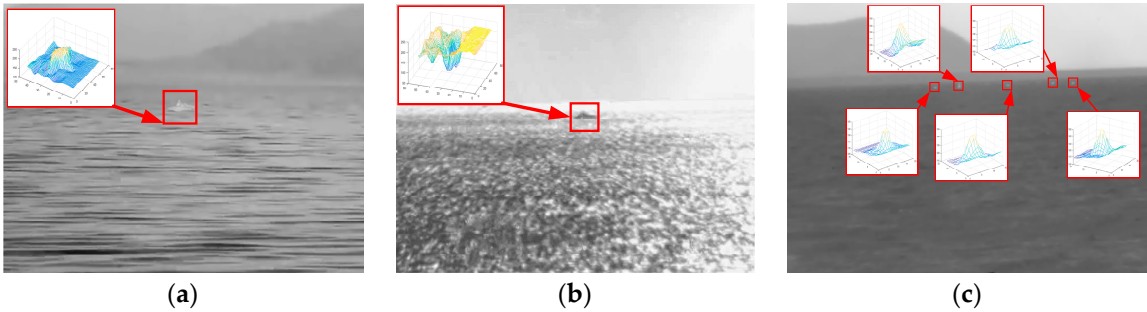

       (**a**)                         (**b**)                        (**c**)

**Figure 2.** The three representative TIR ship images with complex sea background clutter. (**a**) Original image with a bright ship target submerged in long ribbon-like sea clutter, (**b**) original image with a dark ship target buried in strongly fluctuant sun-glint clutter, (**c**) original image with multiple bright point ship targets that appeared near the sea-sky line/coastline.

*2.2. Gray-Level Morphological Reconstruction for Sea Clutter Suppression and Saliency Detection*

2.2.1. Application of Classical Mathematical Morphology in TIR Image Processing

    Mathematical morphology has become a well-known non-linear analysis tool in the field of digital image processing due to its highly parallel processing properties and structured element set theory [34]. In general, the mathematical morphology is constructed with two parts: Structuring element and two basic morphological operators (dilation and erosion). In grayscale TIR image, given the original image $f$ and the selected structuring element $B$, the two basic morphological operators of $f(x, y)$ by $B(m, n)$ are defined as follows:

$$f \oplus B(x, y) = \sup_{\substack{m, n \in D_b \\ x, y \in D_f}} f(x + m, y + n), \tag{2}$$

$$f \Theta B(x, y) = \inf_{\substack{m, n \in D_b \\ x, y \in D_f}} f(x + m, y + n), \tag{3}$$

here, the $f \oplus B(x, y)$ and $f \Theta B(x, y)$ are named as dilation and erosion operator of $f$ at coordinate pixel $(x, y)$, respectively. $D_f$ is the domain of $f$, and $D_B$ is the domain of $B$. In our experiments, $B$ is selected as a 15-pixel disk-shaped structuring element. The dilation (erosion) operator makes the image intensity bigger (smaller) than the original infrared image by using the supremum (infimum) operation. Based on the two basic morphological operators, the definitions of the opening and closing operation of the original image are expressed as:

$$f \circ B(x, y) = (f \Theta B) \oplus B(x, y), \tag{4}$$

$$f \bullet B(x, y) = (f \oplus B) \Theta B(x, y), \tag{5}$$

where the $f \circ B$ is the opening operation and the $f \bullet B$ is the closing operation of $f$. The opening and closing operators can separately remove the bright and dark regions that are smaller than the selected structuring element $B$. Therefore, the opening (closing) operators can be used to approximately forecast the backgrounds in the infrared bright (dark) ship target image $f$. Then, the classical top-hat filter and bot-hat filter of $f$, expressed as $THF$ and $BHF$, are defined as:

$$THF(x, y) = f(x, y) - f \circ B(x, y), \tag{6}$$

$$BHF(x, y) = f \bullet B(x, y) - f(x, y), \tag{7}$$

therefore, the $THF$ extracts bright interesting regions and $BHF$ highlights dark interesting regions in image $f$. In TIR ship image, the ship target is usually a bright (dark) region, so THF/BHF can be directly used to detect ship targets [6,35]. For infrared ship image with high SCR, the classical THF/BHF filters can achieve good detection performance. However, for low SCR, where the ship target is dim and the sea clutter is heavy, the classical THF/BHF filters may suffer serious detection performance degradation, as presented in Figure 3d–f.

Figure 3 shows the processing results of classical THF/BHF filters on three representative TIR ship images with complex sea background clutter. The first row are the original TIR images. Figure 3d,f are the results of Figure 3a,c directly filtered by classical THF, and Figure 3e is the result of Figure 3b filtered by classical BHF. It can be seen from Figure 3d–f that classical morphological filter is robust to the horizon line, island and sky region, but cannot sufficiently enhance weak ship target signal under heavy sea clutter in three images. This is because the classical morphological filters simply adopt local minima or maxima operation in selected structuring element to estimate background without appropriately considering the differences between the target and surrounding sea clutter regions and cannot effectively suppress sea clutter in the uneven maritime scenes. To overcome the shortcomings of classical morphological filters and to accurately detect ship targets buried in heavy sea clutter, a robust ship target detection method based on morphological reconstruction and multi-feature analysis is proposed by fully considering the TIR imaging properties between ship targets and sea clutter.

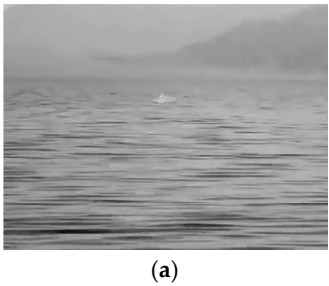 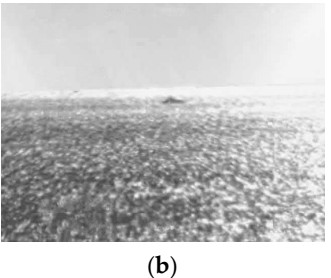 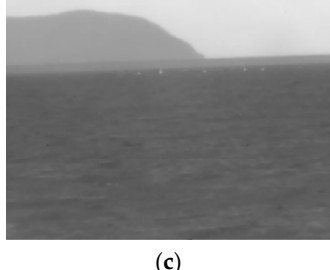

(**a**)  (**b**)  (**c**)

**Figure 3.** *Cont.*

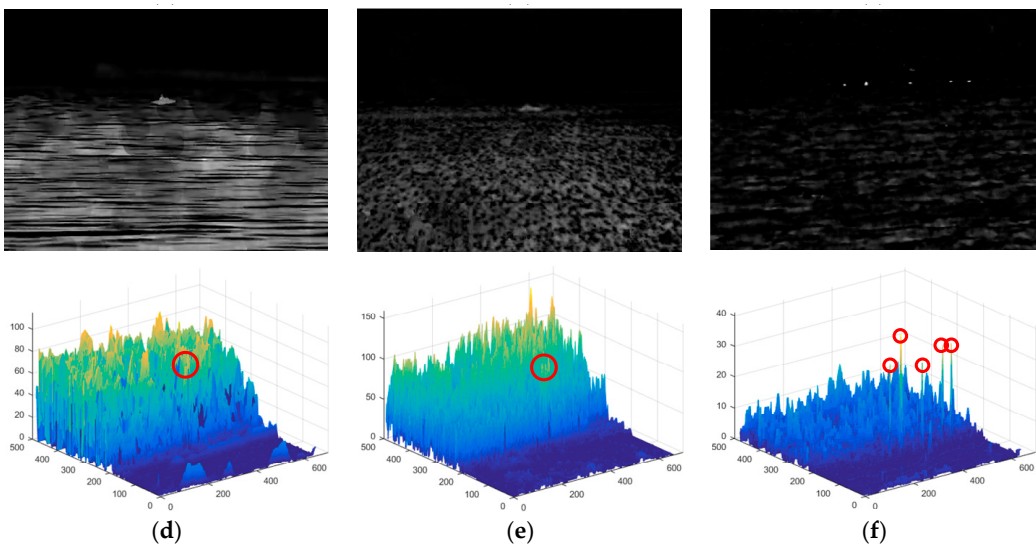

**Figure 3.** The processing results of classical THF/BHF filters on three representative TIR ship images with complex sea background clutter. (**a**–**c**) Original TIR images, (**d**) the result of (**a**) filtered by classical THF. (**e**) The result of (**b**) filtered by BHF. (**f**) The result of (**c**) filtered by THF.

### 2.2.2. Pre-Processing and Intensity Foreground Saliency Detection

References [36,37] indicated that gray-level morphological reconstruction (GMR) can be used to extract objects, which are connected components with the same intensity value and larger (smaller) than the intensity value of the external boundary pixels, while keeping their intensity, shape, and contour detail information and suppressing trivial background clutter and noise. Motivated by this vision, as analyzed in above Section 2.1 part, since the ship target is also a small uniform region with higher (lower) contrast compared with its surrounding backgrounds in TIR remote sensing image, we adopt morphological reconstruction [38] on gray-level infrared images in a dual method to highlight ship targets and remove sea clutter simultaneously. Concisely, we let the result of the opening operation $g = f \circ B$ as the marker image, original TIR image $f$ as the mask image, hence the opening-based gray-level morphological reconstruction (OGMR) of $f$ from $g$, denoted by $OGMR_f[g]$, is computed by iterating elementary geodesic dilations of $g$ under $f$ until stability is reached:

$$OGMR_f[g](x,y) = \underset{i \geq 1}{\cup} gd^{(i)}(x,y), \tag{8}$$

where $gd^{(i)}(x,y)$ is the *i*-th iterating elementary geodesic dilation and the geodesic dilation is defined as follows:

$$gd^{(i)}(x,y) = [g \oplus B] \cap f(x,y) = \min[g \oplus B(x,y), f(x,y)], \tag{9}$$

where intersection operation $\cap$ stands for pixel-wise minimum and union operation $\cup$ stands for pixel-wise maximum. OGMR attempts to automatically restore the original image $f$ from the result image of the opening operation $g$ by using an iterative process, and then the connected regions brighter than the surrounding backgrounds in $f$ will be fully removed, while the dark connected regions will be restored completely with a fine shape-preserving capability, as illustrated in Figure 4c. Therefore, considering the good performances of OGMR, the OGMR operation is used for pre-processing TIR dark ship image in this paper to enhance ship targets and suppress heavy sea clutter and noise. In addition, because ship target and sea clutter have different thermal radiation characteristics, the OGMR can also be used to roughly estimate the background of infrared bright ship image, and then the proposed intensity foreground saliency map (IFSM) for infrared bright ship image is defined as follows:

$$IFSM_b(x,y) = \|f(x,y) - OGMR_f[g](x,y)\|^2, \tag{10}$$

where $\|\cdot\|$ represents $\ell_2$-norm.

Similarly, let $h = f \bullet B$, the closing-based gray-level morphological reconstruction (CGMR) of $f$ from $h$, denoted as $CGMR_f[h]$, is computed by iterating elementary geodesic erosions $h$ above $f$ until stability is achieved:

$$CGMR_f[h](x,y) = \bigcap_{i \geq 1} ge^{(i)}(x,y), \tag{11}$$

where $ge^{(i)}(x,y)$ is the $i$-th iterating elementary geodesic erosion, defined as:

$$ge^{(i)}(x,y) = [h \ominus B] \cup f(x,y) = \max[h \ominus B(x,y), f(x,y)], \tag{12}$$

CGMR tries to automatically restore the original image $f$ from the result image of closing operation $h$ by using an iterative process, and then the connected regions darker than the surrounding backgrounds in $f$ will be fully removed, while the bright connected regions will be restored completely with a fine shape-preserving capability, as Figure 4b demonstrates. Accordingly, CGMR operation can be used for pre-processing infrared bright ship image to enhance a ship target and suppress sea clutter. Meanwhile, the CGMR can also be used to roughly estimate the background of infrared dark ship image, and then the IFSM for dark ship image is derived as:

$$IFSM_d(x,y) = \|f(x,y) - CGMR_f[h](x,y)\|^2. \tag{13}$$

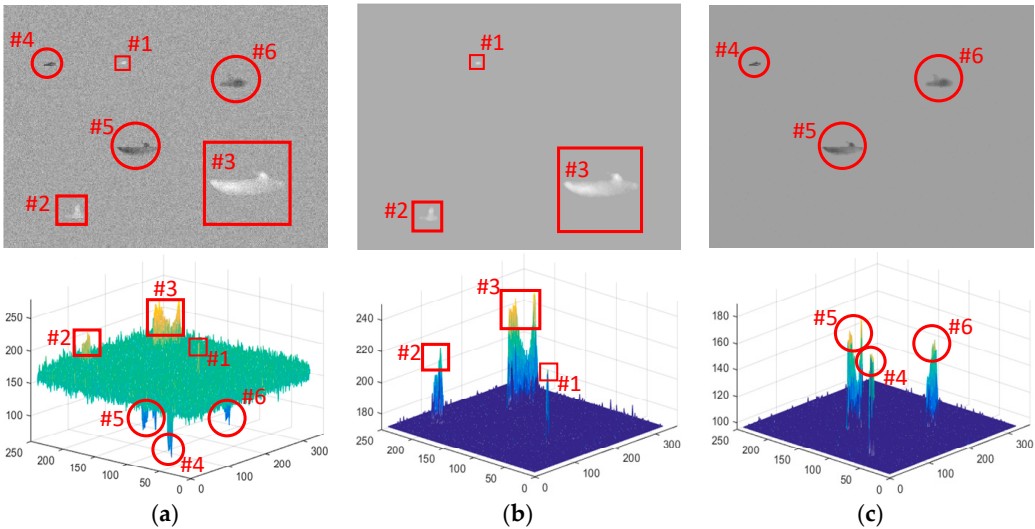

**Figure 4.** The illustration of pre-processing results of synthetic TIR ship image based on gray-level morphological reconstruction (GMR). (**a**) Original synthetic TIR image that contains three bright ship targets and three dark ship targets submerged in heavy white Gaussian noise at the standard deviation of 10. (**b**) The pre-processing result for bright ship target image filtered by closing-based gray-level morphological reconstruction (CGMR). (**c**) The pre-processing result for dark ship target image filtered by opening based gray-level morphological reconstruction (OGMR).

Figure 4 shows the illustration of pre-processing results of a synthetic TIR ship image based on opening or closing GMR. Their corresponding 3-D mesh plots are shown in the second row. The bright ship targets are labeled in boxes and the dark ship targets are labeled in circles. Figure 4a is the original synthetic TIR image including three bright ship targets and three dark ship targets, which are polluted by heavy white Gaussian noise at the standard deviation of 10. Figure 4b is the pre-processing result for bright ship target image filtered by CGMR, and Figure 4c is the pre-processing result for dark ship target image filtered by OGMR. The average pixel intensity of the background in Figure 4a–c is 165, 176, and 158, respectively. It can be seen from Figure 4b,c that after the simulated TIR ship target image filtered by CGMR (OGMR), the dark (bright) components, and heavy noise clutter are totally removed

and the bright (dark) ship targets are completely restored with almost no loss of important signals. Therefore, because the synthetic TIR image contains various types of ship targets and heavy noise, the simulation results show that the OGMR- or CGMR-based pre-processing not only is robust to ship targets with different quantities, sizes, and shapes contaminated by heavy noise clutter, but also has fine intensity, shape, and contour-preserving capability.

To further illustrate the efficiency and robustness of the proposed method, we conducted experiments on the three selected representative TIR ship images with complex sea scene clutter. Figure 5 shows their corresponding pre-processing results for sea clutter removal based on GMR. Figure 5d,f shows the pre-processing results of Figure 5a,c based on CGMR, respectively. Figure 5e shows the pre-processing result of Figure 5b based on OGMR. By comparing Figure 5a–c with Figure 5d–f, the sea clutter can be effectively removed and the detailed information of ship targets can be completely retained by the GMR-based pre-processing. This benefits largely from the fact that opening or closing marker-controlled GMR is a robust and flexible approach for removing local maxima or minima regions of sea clutter with closed contours, where their boundaries are depicted as ridges. Meanwhile, the opening or closing marker is used to reconstruct the original TIR image, so each object including bright or dark ship targets on the desired ridges can be automatically separated to preserve their boundaries.

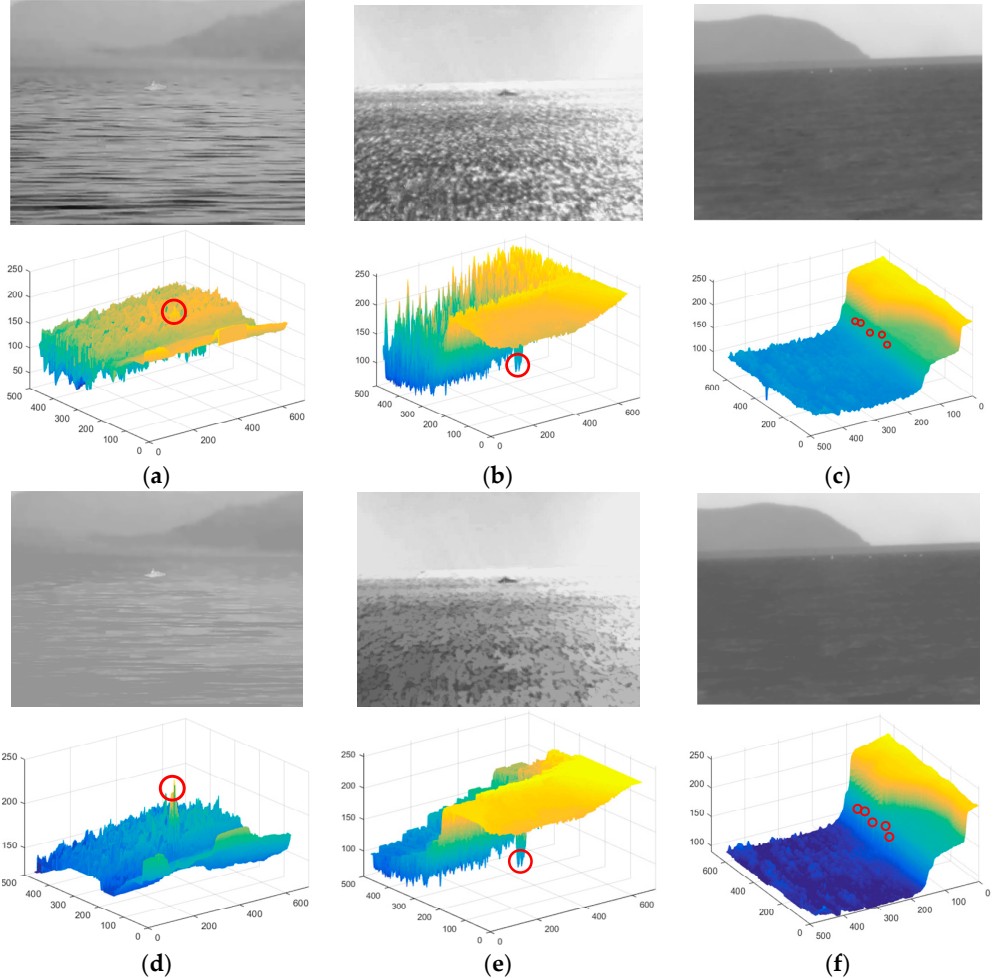

**Figure 5.** The pre-processing results for removing sea clutter based on gray-level morphological reconstruction. (**a**–**c**) are selected three representative TIR ship images with complex sea background clutter, (**d**) the pre-processing result of (**a**) filtered by CGMR, (**e**) the pre-processing result of (**b**) filtered by OGMR, (**f**) the pre-processing result of (**c**) filtered by CGMR.

Figure 6 shows the processed results of TIR ship images by IFSM. The estimated backgrounds of original TIR ship images by OGMR or CGMR are shown in the first row of Figure 6. Figure 6d,f are the results of Figure 5a,c processed by proposed IFSM for bright ship image, and Figure 6e is the result of Figure 5b processed by IFSM for dark ship image. As can be seen from Figure 6d–f, for these three complex maritime scenes, the IFSM can greatly suppress most sea clutter and highlight the ship targets at the same time. This is because the OGMR or CGMR uses region merging and iterative approach method to automatically estimate the gray level of backgrounds, just like a boundary constrained region merging method, as presented in Figure 6a–c. Then according to the distinctive gray intensity characteristics between ship targets and sea background clutter in TIR image, the square of the difference between the original image and the estimated background is used to highlight the intensity foreground saliency map. Therefore, even for chaotic sea clutter, the GMR can estimate the local gray level of the background and IFSM can well highlight the foreground map of potential ship targets.

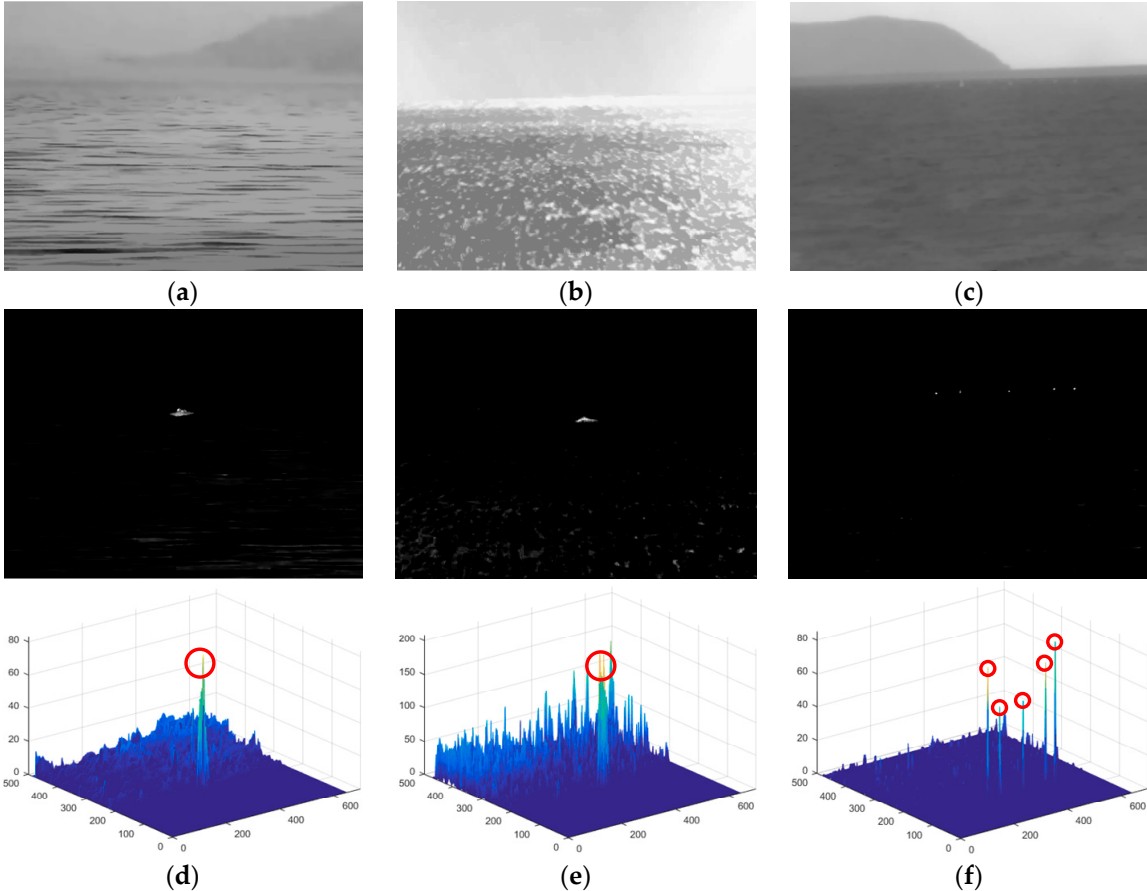

**Figure 6.** The processed results of TIR ship images by intensity foreground saliency map (IFSM). (**a**) The estimated background of Figure 5a by OGMR, (**e**) the estimated background of Figure 5b by CGMR, (**f**) the estimated background of Figure 5c by OGMR, (**d**) the result of (**a**) processed by $IFSM_b$, (**e**) the result of (**b**) processed by $IFSM_d$, (**f**) the result of (**c**) processed by $IFSM_b$.

### 2.2.3. Brightness Contrast Saliency Detection

As can be seen from Figure 5d–f, the pre-processed images based on OGMR or CGMR can efficiently remove dense and intricate sea clutter while completely preserving the ship target signals including intensity, shape, and contour information. Therefore, considering the favorable performances of OGMR or CGMR pre-processed images, we try to fully exploit the brightness uniformity and the local region contrast of TIR ship target to further enhance ship target and suppress trivial clutter. Firstly,

similar to cascade morphological operation, the opening or closing operation with selected structuring element *B* is employed on the pre-processed TIR ship image obtained by CGMR or OGMR to perceive trivial background clutter. Then, the proposed brightness contrast saliency map (BCSM) for bright (dark) infrared ship image is defined as follows:

$$BCSM_b(x,y) = CGMR_f[h](x,y) - CGMR_f[h] \circ B(x,y), \tag{14}$$

$$BCSM_d(x,y) = OGMR_f[g] \bullet B(x,y) - OGMR_f[g](x,y), \tag{15}$$

By the definition of BCSM, the BCSM can be simply treated as two steps: The first step utilizes CGMR or OGMR to efficiently remove heavy sea clutter according to region–neighbor contrast, and the second step uses opening or closing operation to perceive trivial background clutter and pixel-wise subtraction to probe bright or dark regions where potential ship targets may exist. Figure 7 shows the processed results of TIR ship images by BCSM. The pre-processed infrared ship images based on CGMR or OGMR are shown in the first row, and the second row displays their acquired BCSM results, respectively. As can be seen by comparing Figures 7d–f and 3d–f, for these three images, the proposed BCSM outperforms the classical morphological filters, because the proposed BCSM algorithm fully exploits the information of the spatially local region contrast and brightness uniformity of the ship targets.

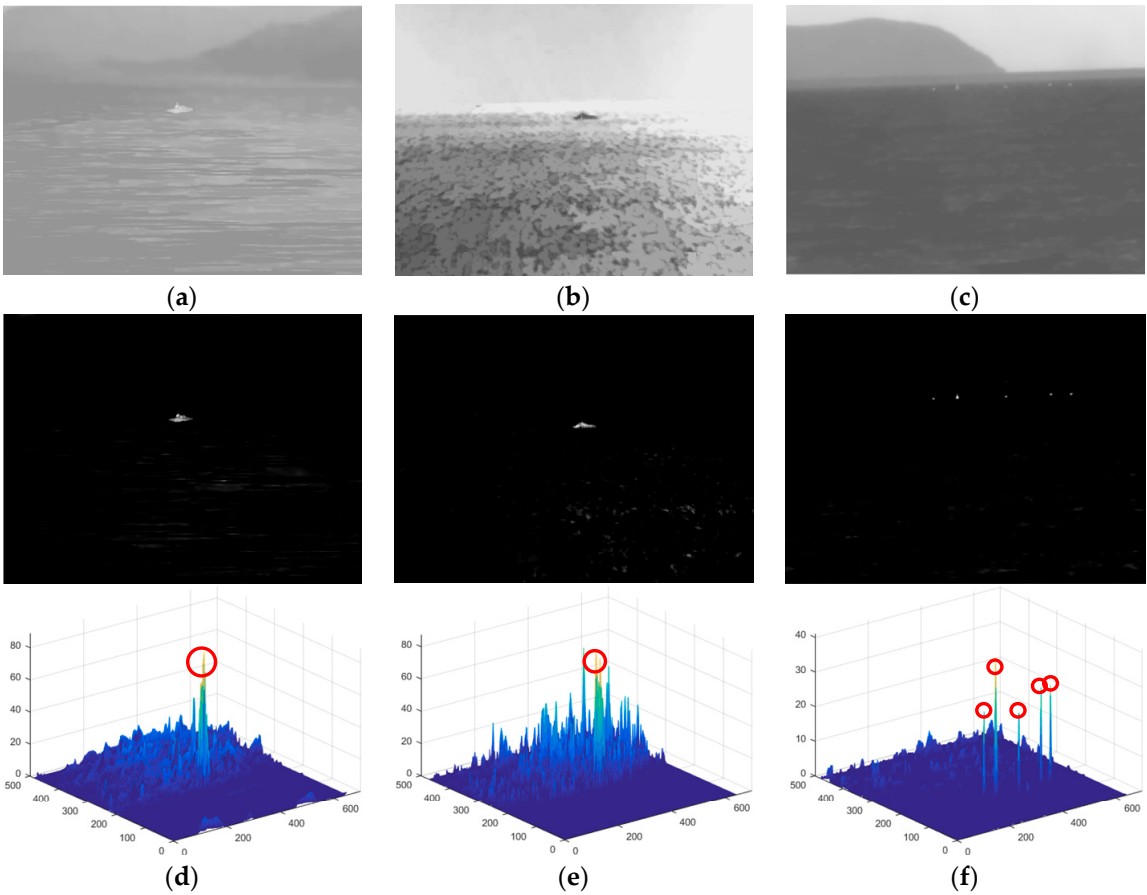

**Figure 7.** The results of TIR ship images processed by brightness contrast saliency map (BCSM). (**a**–**c**) are pre-processed TIR images by gray-level morphological reconstruction, (**d**) the result of (**a**) processed by $BCSM_b$, (**e**) the result of (**b**) processed by $BCSM_d$, (**f**) the result of (**c**) processed by $BCSM_b$.

### 2.3. Saliency Map Fusion Based on Gray-Level Morphological Reconstruction

The previous part has discussed in detail the computation process of the intensity foreground saliency detection and brightness contrast saliency detection based on gray-level morphological reconstruction, respectively. From the above mentioned, the IFSM depicts the distinctive gray intensity features between TIR ship targets and sea background clutter. The BCSM describes the spatial properties of TIR ship targets against background clutter, including brightness uniformity and local region contrast features. The purpose of saliency map fusion is to improve the ship targets' reliability in an infrared image and reduce false alarms by utilizing the information of all feature saliency maps. However, because the above two saliency maps have different magnitudes, we should normalize all the saliency maps to the same range [0, 1] by maximum and minimum values of each map. Given a saliency map $S$, its normalization operation can be expressed as:

$$\mathbf{N}[S](x, y) = \frac{S(x, y) - S_{\min}}{S_{\max} - S_{\min}}, \tag{16}$$

where $S_{\max}$ and $S_{\min}$ are the maximum and minimum values of saliency map $S$. $\mathbf{N}[S](x, y)$ denotes the normalized result of $S$ at location $(x, y)$. Accordingly, the normalized result of IFSM can be written as $\mathbf{N}[IFSM]$, and the normalized result of BCSM can be written as $\mathbf{N}[BCSM]$. Therefore, the proposed final saliency map is generated by a pixel-wise multiplication fusion manner:

$$SM_b(x, y) = \mathbf{N}[IFSM_b](x, y) \times \mathbf{N}[BCSM_b](x, y), \tag{17}$$

$$SM_d(x, y) = \mathbf{N}[IFSM_d](x, y) \times \mathbf{N}[BCSM_d](x, y), \tag{18}$$

here, $SM_b$ and $SM_d$ represent the final saliency maps of infrared bright and dark ship images, respectively. Then, the candidate ship targets can be separated and extracted from the final saliency map $SM$ by an adaptive threshold [4,24], and the adaptive threshold is determined as:

$$\tau = \overline{SM} + \varepsilon \times \sigma(SM), \tag{19}$$

where $\overline{SM}$ and $\sigma(SM)$ are the mean value and standard deviation of the final saliency map $SM$, respectively. $\varepsilon$ is an experimentally selected constant and it can be chosen from the interval [10,15] for most scenarios. Finally, the binary segmentation result $BSM$ of the final saliency map is acquired:

$$BSM(x, y) = \begin{cases} 1, & SM(x, y) \geq \tau \\ 0, & otherwise \end{cases}, \tag{20}$$

Figure 8 gives the final results of saliency map fusion. The first row is the results of saliency map fusion of Figure 5a–c, their corresponding 3-D mesh plots are shown in the second row, and the third row is the binary segmentation result $BSM$ of their final saliency maps. The red boxes have enlarged point ship targets, and the purple boxes have enlarged small-size false targets. As shown in Figure 8a,c, the proposed saliency map can be directly used for detecting TIR ship targets with different sizes and quantities submerged in complex backgrounds with sea-sky line/coastline or long ribbon-like sea clutter. Figure 8b demonstrates that for low-contrast ship targets buried in intricate backgrounds with strongly rolling sun-glint clutter, the proposed saliency map can make ship targets stand out but fail to distinguish real ship targets from chaotic sea clutter, because some strong clutter also has non-negligible saliency even in the final saliency map. To cope with this problem, a two-step ship verification strategy is introduced in the following Section 2.4 part based on contour feature description and shape feature constraint.

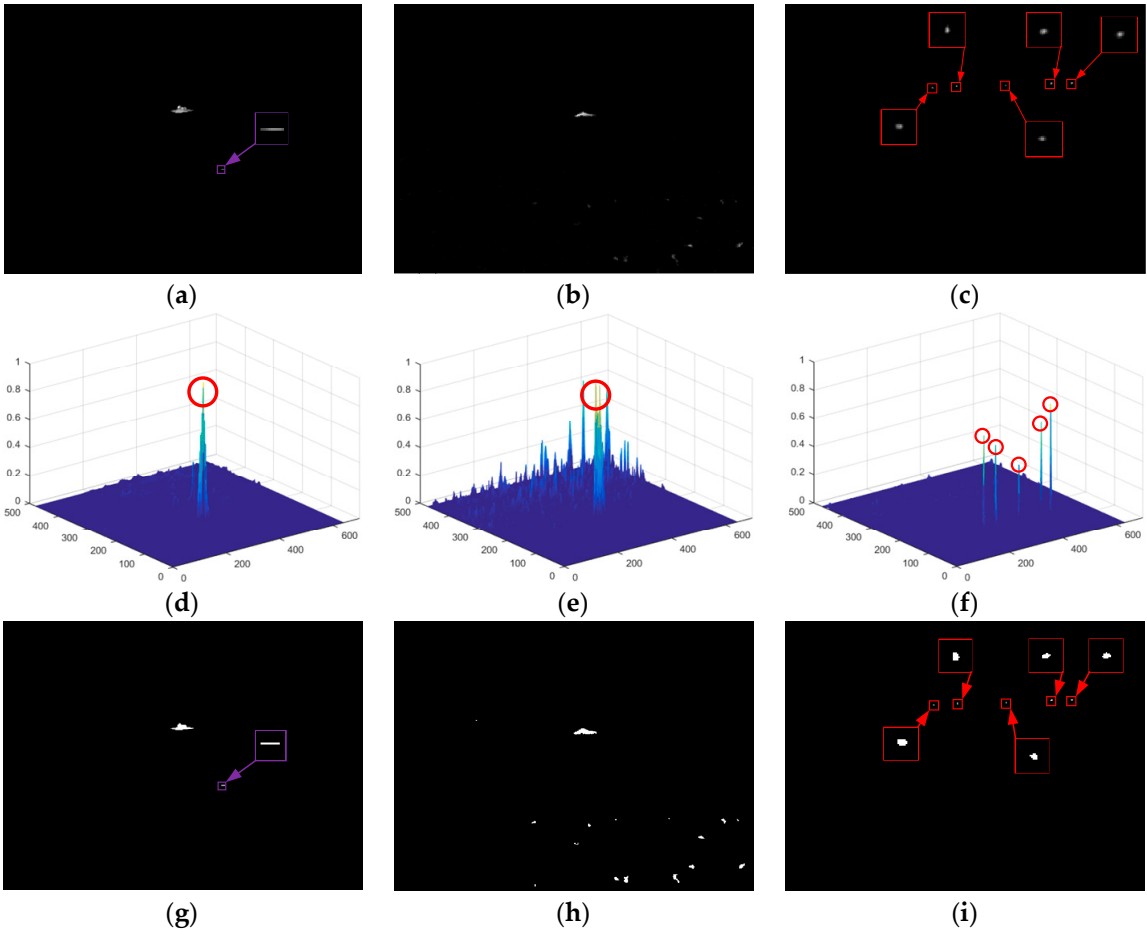

**Figure 8.** The final results of saliency map fusion. (**a**–**c**) are the final results of saliency map fusion of Figure 5a–c, (**d**–**f**) are their corresponding 3-D mesh plots, (**g**–**i**) the binary segmentation results of their final saliency maps.

### 2.4. Ship Target Verification Based on Contour Description and Shape Constraint

#### 2.4.1. Contour Description of TIR Ship Based on Eigenvalue Analysis of Structure Tensor

For TIR remote sensing images, the pixel intensities of small ship targets tend to be evenly distributed, they have an appearance of high uniformity and will not show large gray scale changes. Therefore, after GMR-based saliency map fusion, the contour information of ship targets can be completely retained. Recall that the contours of ship target in the OGMR and CGMR preprocessed images, as shown in Figure 5d–f, are depicted as ridges, like a simplified level-set model. Meanwhile, inspired by the successful applications of closed contour extraction, such as Chan–Vese model in ship detection [20,21], a novel contour measure based on GMR and eigenvalue analysis of structure tensor is presented to characterize candidate ship targets and eliminate residual clutter simultaneously. Structure tensor has become a powerful tool for edge analysis, and achieved some impressive results for its good perceptivity on the dominant direction at the local image [39,40]. To investigate the local contour information of TIR ship targets, the structure tensor *ST* is firstly calculated as:

$$ST(x,y) = G_\sigma * \left\{ \nabla u(x,y) \times [\nabla u(x,y)]^T \right\} = G_\sigma * \begin{bmatrix} \left(\frac{\partial u}{\partial x}\right)^2 & \frac{\partial^2 u}{\partial x \partial y} \\ \frac{\partial^2 u}{\partial x \partial y} & \left(\frac{\partial u}{\partial y}\right)^2 \end{bmatrix} = \begin{bmatrix} ST_{11} & ST_{12} \\ ST_{21} & ST_{22} \end{bmatrix}, \tag{21}$$

where $G_\sigma$ is a Gaussian kernel with variance σ and $*$ is a convolution operator, *u* denotes pre-processed infrared ship image after OGMR or CGMR operation. The Gaussian kernel can be considered as

a window of *ST* to perceive the local information and σ determines the size of the window. The σ is empirically set to 2. Then the normalized large eigenvalue of matrix *ST* is obtained:

$$\lambda_{\text{large}}(x, y) = \mathbf{N}\left[\frac{1}{2}\left(ST_{11} + ST_{22} + \sqrt{(ST_{11} - ST_{22})^2 + 4ST_{12}ST_{21}}\right)\right], \tag{22}$$

here, we use normalized large eigenvalue $\lambda_{\text{large}}$ of structure tensor to delineate the edges of an infrared ship image. Because the large eigenvalue of structure tensor can indicate the predominate direction and the coherence degree of the gradient trend, the eigenvalues $\lambda_{\text{large}}$ of the closely adjacent surroundings of infrared ship targets are almost the maximum values in the global eigenvalue image and present an approximately closed contour. Based on this cue, the average eigenvalue measure of structure tensor (STAEM) is proposed to characterize the contour feature of candidate TIR ship targets and eliminate residual clutter simultaneously, and the STAEM can be defined as:

$$\lambda_{aver} = \frac{\sum\limits_{(m,n)\in D_{con}} \lambda_{\text{large}}(m, n)}{\sum\limits_{(m,n)\in D_{con}} 1}, \tag{23}$$

where $\lambda_{aver}$ denotes the STAEM value of the specified connected region. $D_{con}$ is the domain of binary contour obtained by dilation of the connected region. On the right side of the formula, the numerator is the total value of normalized large eigenvalues in the domain $D_{con}$, and the denominator indicates the total pixel number of the domain $D_{con}$. Finally, any connected regions in the binarized final saliency map whose STAEM value is larger than a distinctive threshold *Thr\** are retained, and all other pixels are set to 0. The threshold *Thr\** is experimentally set to 0.2094 for most scenarios, and the threshold selection will be discussed in detail in the Effects of Parameters part. Figure 9 illustrates the calculation process of STAEM of a connected region. Figure 9a is the computed contour of the connected region by grouping normalized large eigenvalues $\lambda_{\text{large}}$, and its binary contour $D_{con}$ is obtained by dilation with three-pixel disk-shaped structuring element, as shown in Figure 9d.

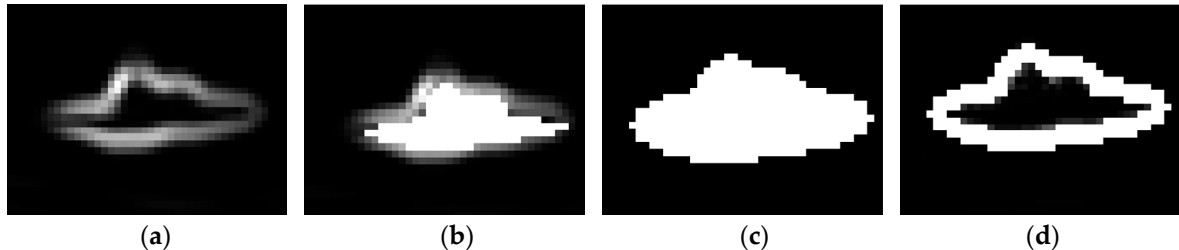

| (a) | (b) | (c) | (d) |

**Figure 9.** The illustration of the calculation process of STAEM around TIR ship targets. (**a**) is the contour computed by grouping normalized large eigenvalues, (**b**) binary ship target and its computed contour, (**c**) the dilated binary ship target, (**d**) binary contour obtained by dilation of ship targets and its true computed contour.

Figure 10 provides the final detection results according to STAEM calculation. The red boxes have enlarged true ship targets, and the purple boxes have enlarged small-size false targets. First row is the normalized large eigenvalue maps of structure tensor of pre-processed images in Figure 5d–f, then, second row displays the binarized final saliency maps and the computed STAEM maps simultaneously, and the third row gives the final detection results after the threshold division of STAEM values. As shown in Figure 10d–f, for these three complex infrared images, the STAEM values of ship targets are much larger than those of sea clutter, hence, the STAEM calculation can be used to distinguish candidate infrared ship targets from residual clutter by thresholding operation. The reason can be illustrated by the crucial differences between infrared ship targets and sea clutter in terms of the thermodynamic character and the imaging procedure. In TIR remote sensing images, the pixel

intensities of small ship targets have high uniformity, and their solid contours appear almost totally closed and have larger eigenvalues in structure tensor map. While for sea clutter, due to their pixel intensities are uneven, their contours present a semi-open form after saliency map fusion based on gray-level morphological reconstruction.

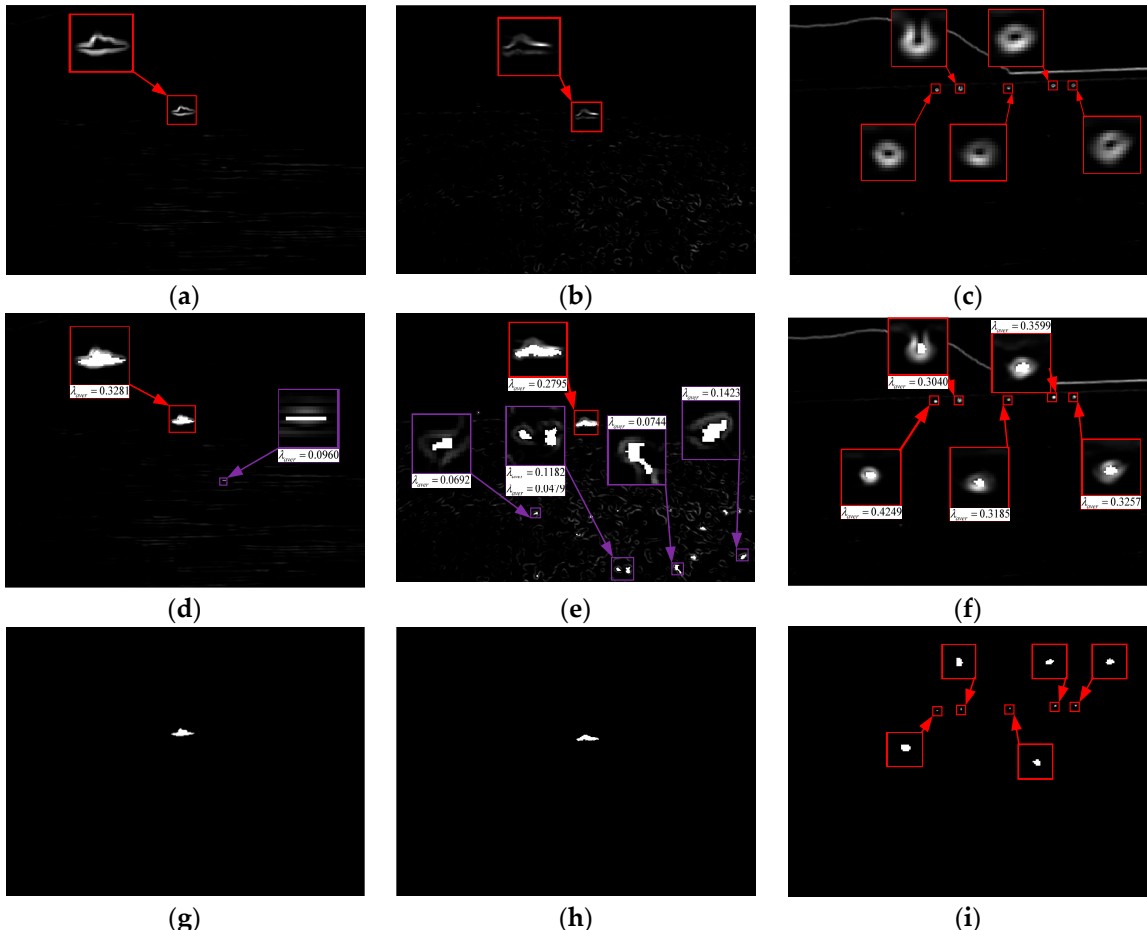

**Figure 10.** The final detection results according to STAEM calculation. (**a**–**c**) are the STAEM maps of Figure 5. (**d**–**f**) display the binarized saliency map fusion and STAEM maps simultaneously, and the STAEM values of true ship targets and some representative false alarms are marked, (**g**–**i**) the final detection results after the threshold division of STAEM values.

2.4.2. Shape Constraint for TIR Ship Identification Based on Statistical Knowledge

After the above GMR induced operation steps, the binarized candidate ship targets are successfully extracted from backgrounds, and their simple geometric shape parameters, such as *Perimeter* and *Area*, are retained and rapidly obtained. The geometric shape feature not only can efficiently distinguish the true ship targets from the non-ship targets, such as ocean waves, tail waves, sea-sky line/coastline, and islands but also is less sensitive to the change of viewing angle and ship target size. As some previous studies [23,41] mentioned that the ship target usually appears as a long narrow body, and the main body of the ship target has a regular shape feature. Hence, three widely used shape descriptors are adopted in this paper:

$$Ratio_{mami} = \frac{L_{majoraxes}}{L_{minoraxes}}, \tag{24}$$

$$Compactness = \frac{(Perimeter)^2}{Area}, \tag{25}$$

$$Rectangularity = \frac{Area}{Rectangle}, \tag{26}$$

where $L_{majoraxes}$ and $L_{minoraxes}$ are the lengths of the major axes and minor axes, respectively. *Rectangle* indicates the circumscribed rectangle of the candidate region. The ratio of major to minor axes $Ratio_{mami}$ can be used to describe its best fit ellipse because ship targets are usually long and thin. The *Compactness* depicts the degree of circular similarity by fully considering the influence of object boundary changes on the average radius. The *Rectangularity* represents the degree of rectangular similarity by assuming the values in the range of [0, 1], and 1 indicates the perfect rectangular region.

Shape feature constraint methods are widely used to identify a ship target in the field of infrared ship target detection [14,23], and those methods roughly set a large constraint range of shape parameters according to the statistics of ship shape parameters in their own datasets to output the final detected ship targets. However, because small ship targets are captured by long-distance TIR imaging sensors, the dynamic range of their shape parameters is relatively small and specific. By analyzing the TIR imaging characteristics of small ships and the statistics and observation of small ships in our datasets, it was found that the main distribution range of their shape parameters is similar to that of most TIR ships in VAIS database [31]. The VAIS database contains 1242 sliced TIR ship images, which are composed of 264 different types of ships captured during daytime and nighttime with variable view-angles and diverse distances. Therefore, in order to further identify true ship targets according to shape features, we randomly select 700 TIR ship targets from VAIS database to compute and count their shape parameters. Through the statistics and observation of the shape parameters of selected TIR ship targets, the statistical shape knowledge is reliably obtained and can be effectively utilized to further distinguish true ship targets from non-ship targets. Figure 11 shows some representative TIR ship targets selected from VAIS database and their corresponding binary images (manually labeled). Table 2 shows the statistical measurements of three shape descriptors on 700 TIR ship targets selected from VAIS database. Additionally, according to the statistical knowledge in Table 2, in order to further accurately identify small ship targets with shape feature, the ranges of $Ratio_{mami}$, *Compactness* and *Rectangularity* in our paper are also empirically set to 1.1002~8.7857, 11.2813~118.6254, and 0.3826~0.8769, respectively. This means that if the shape parameters of the candidate region are within these ranges, it will be regarded as ship target to be output, and if its parameters are not within these ranges, it will be regarded as a non-ship target to be discarded. The three shape parameters of small ship target in Figure 10g are 2.8824, 27.6935, and 0.4900, respectively. The three shape parameters of small ship target in Figure 10h are 4.1144, 31.9385, and 0.5591, respectively. The shape parameters of these two ship targets are within the ranges of Table 2, so they will be output as the final detected ship targets. Note that point ship targets whose area is smaller than 30 pixels do not have obvious shape information, thus the statistical shape knowledge constraint will not be applied in the identification of ship target if their area is less than 30 pixels. Therefore, the dim point ship targets shown in Figure 10i will also be output as the detected ship targets.

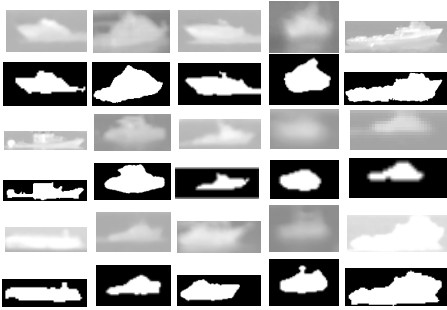

**Figure 11.** The illustration of some selected TIR ship targets from visible and infrared ships (VAIS) database and their corresponding binary images.

**Table 2.** The statistical measurements of three shape descriptors of 700 TIR ship targets selected from VAIS database.

| Measure | Min | Average | Max |
|---|---|---|---|
| $Ratio_{mami}$ | 1.1002 | 3.5105 | 8.7857 |
| $Compactness$ | 11.2813 | 48.1344 | 118.6254 |
| $Rectangularity$ | 0.3826 | 0.6025 | 0.8769 |

## 3. Ship Target Detection based on Morphological Reconstruction and Multi-feature Analysis

### 3.1. Proposed Bright Ship Target Detection in TIR Images

Most studies [11,23,25] indicated that the background is the comparatively dark sea surface and the ship targets might be relatively local brighter regions. According to these above-mentioned methods and theories, the accurate bright ship target detection algorithm for TIR image is established. Firstly, we perform the opening operation on the original TIR image and reconstruct the original image from the result image after the opening operation by OGMR. Therefore, the background of the bright ship target image is estimated by OGMR, and then the IFSM is computed and normalized. Meanwhile, we perform the closing operation on the original image, and reconstruct the original image from the image after closing operation by CGMR. Accordingly, the pre-processing result of bright ship image is obtained by CGMR, and then the BCSM is calculated and normalized. Next, the final saliency map is generated by fusing normalized IFSM and normalized BCSM in a pixel-wise multiplication fusion manner, and the binary segmentation result of the final saliency map is acquired by an adaptive threshold. The whole of the proposed saliency map fusion algorithm for TIR bright ship image based on GMR is summarized in Algorithm 1. It is noteworthy that the Algorithm 1 not only can be used to highlight the potential bright ship targets buried in extremely serious rolling sun-glint clutter, but also can be used to directly detect the ship targets submerged in common sea clutter.

---

**Algorithm 1.** Proposed saliency map fusion algorithm for TIR bright ship image

---

Input: TIR ship target image $f$, structuring element $B$.
Output: Binary final saliency map for bright ship target image $BSM$.
Step 1: Perform the opening operation with structuring element $B$ on $f$ to acquire $g$ using (4).
Step 2: Compute the elementary geodesic dilation $gd^{(i)}$ according to (9).
Step 3: Reconstruct $f$ from $g$ based on OGMR $OR_f[g]$ derived from (8).
Step 4: Obtain the intensity foreground saliency map $IFSM_b$ of $f$ according to (10)
Step 5: Normalize the intensity foreground saliency map $\mathbf{N}[IFSM_b]$ using (16).
Step 6: Perform the closing operation on $f$ to acquire $h$ using (5).
Step 7: Compute the elementary geodesic erosion $ge^{(i)}$ according to (12).
Step 8: Acquire the pre-processing result by reconstructing $f$ from $h$ based on CGMR $OR_f[h]$ derived from (11).
Step 9: Calculate the brightness contrast saliency map $BCSM_b$ on pre-processed image $OR_f[h]$ using (14).
Step 10: Normalize the brightness contrast saliency map $\mathbf{N}[BCSM_b]$ according to (16).
Step 11: Generate the final saliency map $SM_b$ based on pixel-wise multiplication fusion manner according to (17).
Step 12: Calculate the adaptive threshold $\tau$ according to (19).
Step 13: Acquire the binary segmentation result $BSM$ of the final saliency map $SM_b$ according to (20)

---

In order to further identify the true ship targets from residual clutter, a reliable two-step ship verification method for TIR ship images is presented. Firstly, we compute the normalized large eigenvalue map of structure tensor of CGMR pre-processed image and calculate the STAEM values of each labeled connected region in the binary segmentation result of the final saliency map. Then, the map of candidate ship targets is obtained by eliminating the connected regions where the STAEM value is smaller than an experimentally distinctive threshold $Thr^*$. Finally, the output detected infrared

ship target map and target position are acquired by excluding the non-ship targets if their computed shape description parameters are not within the ranges of Table 2. The whole of the proposed two-step ship verification strategy for TIR bright ship image based on contour feature representation and shape feature constraint is summarized in Algorithm 2.

---

**Algorithm 2.** Proposed two-step ship verification strategy for TIR ship image

---

Input: Pre-processed image $u$, binary final saliency map for ship target image *BSM*, and the distinctive threshold $Thr^*$.

Output: Detected TIR ship target map.

Step 1: Compute the regularized structure tensor *ST* of pre-processed image $u$ according to (21).

Step 2: Calculate the normalized large eigenvalue map $\lambda_{\text{large}}$ of matrix *ST* according to (22).

Step 3: Label the connected regions of *BSM* with numbers, and obtain the total number *TN* of connected regions in *BSM*.

Step 4: **for** label index $k = 1{:}TN$ **do**

　　　Compute the average eigenvalue measure of structure tensor (STAEM) $\lambda_{aver}$ of $k$-th connected region according to (23).

　　　Eliminate the $k$-th connected region if the $\lambda_{aver}$ is smaller than distinctive threshold $Thr^*$.

　　　**end for**

Step 5: Obtain the map of candidate ship targets and acquire the total number *SN* of candidate ship targets.

Step 6: **for** candidate ship index $t = 1{:}SN$ **do**

　　　Compute the shape description parameters of $t$-th candidate region according to (24)–(26).

　　　Exclude the non-ship targets if their computed shape description parameters are not within the ranges of Table 2.

　　　**end for**

Step 7: Acquire the output detected ship target map and target position

---

### 3.2. Proposed Small Ship Target Detection in TIR Images

Currently, studies mostly focus on bright infrared ship target detection. However, in real cases, dark ship targets whose infrared radiation is lower than surroundings also exist in the backlighting infrared images. To detect dark infrared ship targets, Equations (4), (9), (8), and (10) in the Steps 1–5 of Algorithm 1 can be replaced by Equations (5), (12), (11), and (13), respectively, to compute normalized IFSM of dark ship image. Meanwhile, Equations (5), (12), (11), and (14) in the Steps 6–10 of Algorithm 1 can be replaced by Equations (4), (9), (8), and (15), respectively, to compute normalized BCSM of dark ship image. Then, the binary segmentation result of the final saliency map of dark ship image is obtained according to the Steps 11–13 of Algorithm 1. Finally, the two-step ship verification strategy according to Algorithm 2 for TIR ship image is adopted to distinguish true ship targets from non-ship targets, and the output detected dark ship target map and target position are reliably acquired.

In the real-world maritime scenes, the local-contrast brightness of ship targets is unknown, so the small ship target detection for both bright and dark ones in infrared image remains to be worthy of further investigation [5,42]. As discussed above, the bright and dark infrared ship target maps can be separately detected by parallel processing algorithms on both sides. The final detected ship target map is obtained by directly adding both bright and dark infrared ship target maps, and the flow chart of the proposed small ship target detection method is illustrated in Figure 1. Therefore, the proposed small ship target detection method has the potential ability for real-time application due to the highly parallel computing properties of multichannel image-processing and mathematical morphology in multicore hardware systems [43,44].

## 4. Experimental Results

In this section, a series of experiments on TIR ship images with various maritime scenes are performed to validate the accuracy and effectiveness of the proposed small ship target detection

algorithm. Furthermore, some classical algorithms and state-of-the-art algorithms are selected for performance comparison.

### 4.1. Test Dataset

The test dataset is composed of 9 TIR maritime sequences, and each sequence represents a typical scenario in TIR ship detection applications. These TIR ship image sequences are labeled as Seq1–9. Figure 12 shows the representative images of the TIR ship image sequences used for performance evaluation. Figure 12a displays a bright ship target located in a relatively mild sea and sky background. Figure 12b represents a bright ship target under a heterogeneous background with sea, sky, and islands. Figure 12c shows a low-contrast bright ship target submerged in long ribbon-like sea clutter. Figure 12d depicts a dark ship target against strong backlighting sea-sky background. Figure 12e is a dark ship target disturbed by huge rolling ocean waves. Figure 12f depicts a low-contrast dark ship target buried in strongly fluctuating sea clutter and sun glints. Figure 12g includes two fast-moving ships with strong long-tail wave interferences [45]. Figure 12h is the case where both bright and dark ship targets appear under heavy sea fog. Figure 12i represents multiple dim bright point ship targets appearing near the sea-sky line/coastline. Table 3 lists the detailed information about the test image sequences. Accordingly, the test images are variable in scene type, clutter type, target type, target shape, and target quantity. Testing on this dataset proves that the algorithm is suitable for many TIR maritime scenarios.

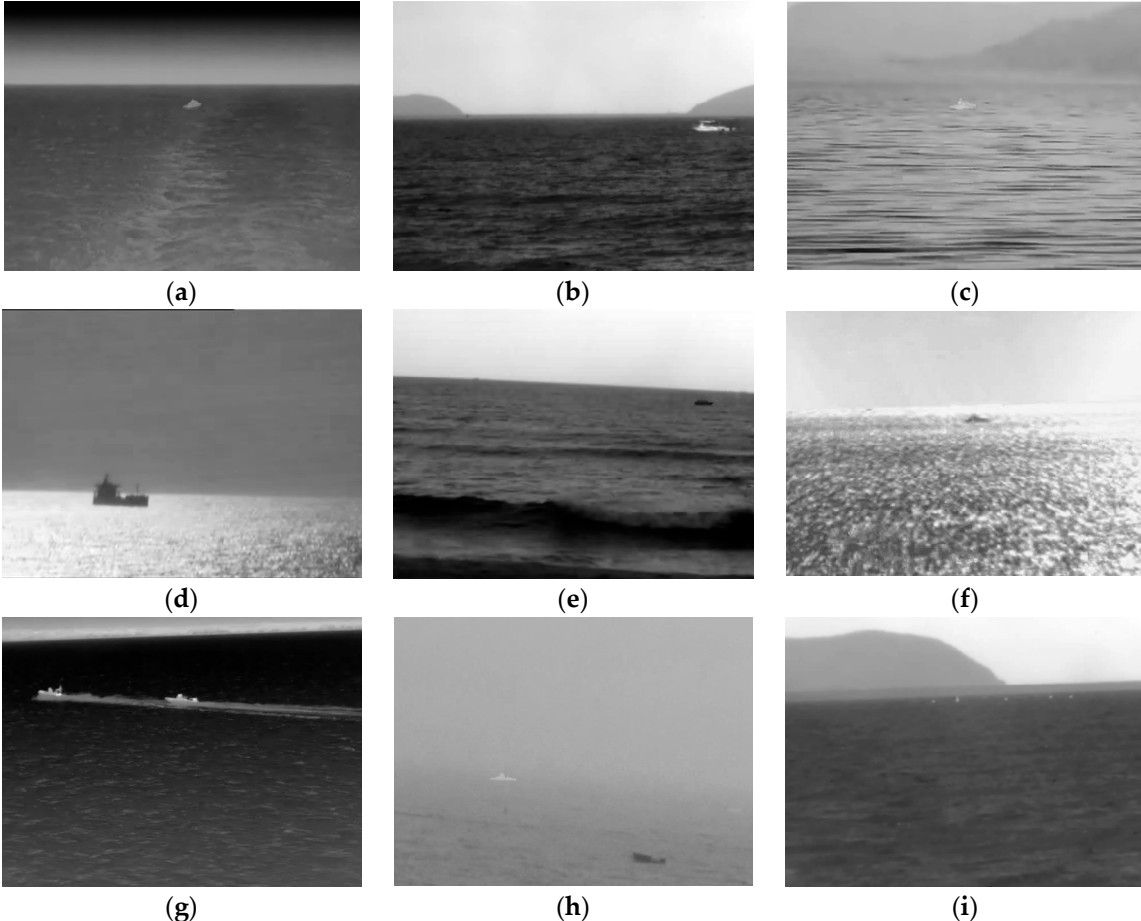

**Figure 12.** The illustration of test images. (**a**) the 25th frame of Seq1, (**b**) the 21st frame of Seq2, (**c**) the 450th frame of Seq3, (**d**) the 12nd frame of Seq4, (**e**) the 112nd frame of Seq5, (**f**) the 301st frame of Seq6, (**g**) the 26th frame of Seq7, (**h**) the 275th frame of Seq8, (**i**) the 181st frame of Seq9.

**Table 3.** The detail information of the test image sequences.

| Sequences | Seq1 | Seq2 | Seq3 | Seq4 | Seq5 | Seq6 | Seq7 | Seq8 | Seq9 |
|---|---|---|---|---|---|---|---|---|---|
| Image size | $640 \times 480$ | $640 \times 480$ | $640 \times 480$ | $320 \times 240$ | $640 \times 480$ | $640 \times 480$ | $640 \times 480$ | $640 \times 480$ | $640 \times 480$ |
| Sea clutter complexity | Medium | Medium | High | Medium | High | High | High | Low | High |
| Target area | 112~308 | 355~513 | 323~487 | 781~1557 | 95~291 | 246~767 | 195~304 | 279~1034 | 6~50 |
| Target brightness | Bright | Bright | Bright | Dark | Dark | Dark | Bright | Bright/dark | Bright |
| Target number | 1 | 1 | 1 | 1 | 1 | 1 | 2 | 2 | 5 |
| Total images | 500 | 500 | 500 | 500 | 500 | 500 | 500 | 500 | 500 |

## 4.2. Results of TIR Ship Detection

### 4.2.1. Effects of Parameters

According to the above Section 2.4.1 part analysis, the STAEM values of ship targets are almost all much larger than those of non-ship targets. Accordingly, in the experiment, we randomly selected 500 ship target blocks and 500 non-ship target blocks and calculated their STAEM values one by one. The calculated STAEM values of the selected blocks are shown in Figure 13. The blue point line denotes the STAEM values of ship targets, and the red point line denotes the STAEM values of non-ship targets. As can be seen from Figure 13, because the STAEM values of ship target blocks and non-ship target blocks have a distinctive distinguishability, the OTSU method [46] is utilized to find the optimal threshold:

$$Thr^* = OTSU(\mathbf{STAEM}), \tag{27}$$

where $\mathbf{STAEM} = [STAEM_1, STAEM_2, \ldots, STAEM_j, \ldots, STAEM_{1000}]$ is the group of STAEM values of selected blocks, and $j$ denotes the index of the selected block. Then, the computed distinctive threshold $Thr^* = 0.2094$ can be used to clearly discriminate ship targets and non-ship targets, as shown by the purple line in Figure 13. Therefore, the distinctive threshold is empirically set $Thr^* = 0.2094$ in our experiments to reliably separate candidate ship targets and non-ship targets.

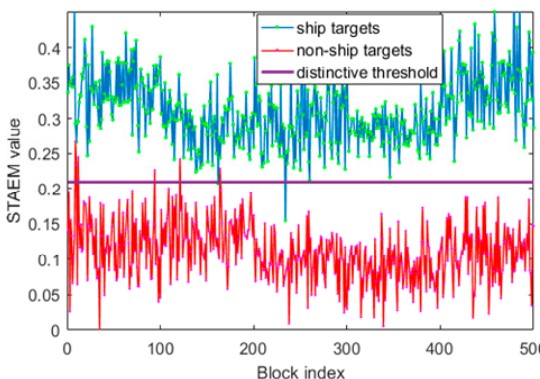

**Figure 13.** The results of the average eigenvalue measure of structure tensor (STAEM) values of the blocks and the experimentally selected distinctive threshold.

### 4.2.2. Visual Comparison to TIR Ship Target Detection Baseline Methods

In this part, four classical ship target detection methods and four state-of-the-art ship target detection methods are introduced in the comparison experiments to evaluate the performance of the proposed TIR ship target detection method. The top-hat/bot-hat filters (THF/BHF) [6,35], 2-D maximum entropy (2DME) [17], 2-D Otsu (2DO) [15] and mean shift segmentation (MSS) [19] are chosen as representative classical ship target detection methods. The iterative multi-feature segmentation (IMFS) [23], improved fuzzy C-means clustering (IFCM) [3], Chan–Vese model (CVM) [21] and histogram cyclic shift transformation (HCST) [13] are selected as representative state-of-the-art methods. Because those methods have been well studied, they can be used for assessing the performance of the new ship target detection method.

Figure 14 shows the ship target detection results of different methods for Figure 12. In the detection result images of Figure 12i, the point ship targets are enlarged for better observation and marked by red boxes. The classical THF/BHF is a commonly used method for background suppression and target enhancement for its simple computation and easy implementation. However, it is vulnerable to heavy clutter and may not perform well for ship targets submerged in heavy sea clutter. It can be seen from Figure 14b1–b9, the detection results of classical THF/BHF can perceive the ship targets but suffer from extremely serious residual clutter. The 2DME and 2DO both are threshold processing methods based on 2-D histogram analysis of the image. When the contrast between the ship target and the background is high, the two methods can be directly used to segment the ship targets. However, these methods are sensitive to the contrast and pixel percentage of ship target and background. Therefore, it can be seen from Figure 14c1–c9,d1–d9 that those two methods obtain the worst detection results in all nine scene images. The MSS is a feature-space analysis algorithm based on graph region merging, so it can efficiently extract the regions of ship targets against a homogenous background. Nevertheless, the MSS does not work well for ship target detection in infrared images with heterogeneous background, as shown in Figure 14e1–e8. More importantly, the MSS method cannot detect point-size ship targets because the point ship targets will be eliminated when regions are merging, as Figure 14e9 illustrates.

The IMFS method reasonably uses two approaches including iterative global thresholding and ship target shape constraint to segment ship targets, hence the IMFS presents better performance than the above four classical methods, as Figure 14f1–f3,f7–f9 shows. The IMFS is based on the assumption that the ship target region is much brighter than the background, so it completely discards the true dark ship targets buried in a brighter sea background, as shown in Figure 14f4–f6. In the IFCM method, the Gaussian filter and top-hat transform are firstly used to smooth background, and then the modified fuzzy C-means clustering (FCM) is applied for ship target segmentation. It uses nonlocal spatial information and spatial shape information effectively to improve the performance of FCM to efficiently suppress noise interference, as depicted in Figure 14g2,g4, g8, g9. However, the top-hat transform and unsupervised fuzzy clustering are both highly sensitive to sea clutter and tail waves, as presented in Figure 14g1,g3,g5–g7. The CVM extracts ship targets through seeking the closed contours of relatively even regions by curve evolution and iterative convex optimization, and it can successfully segment ship targets with clear contour information, as presented in Figure 14h1,h4,h7,h8. Whereas, because heavy clutter can destroy the topology structure of ship targets and the islands may also have strong contour information, the detection results of CVM method will be greatly affected by heavy sea clutter and islands, as Figure 14h2,h3,h5,h6,h9 demonstrates. The HCST is a non-linear histogram curve transformation for maritime target detection based on background modeling, and the effect is excellent for targets with obvious local contrast, as depicted in Figure 14i4,i7–i9. However, for the environmental maritime conditions with multiple background types, the HCST method obtains poor detection performance, as presented in Figure 14i1–i3,i5,i6.

It is noteworthy that those compared methods all fail to detect ship targets for Figure 14a3,a5,a6,a8,a9. Nevertheless, from Figure 14j1–j9, it can be seen that the proposed method can precisely detect all the ship targets with lowest false alarms and is more robust than other compared eight methods. This robustness is attributed to the dual approach with reasonable integration of multiple features after gray-level morphological reconstruction, including intensity, local contrast, contour, and shape features. Considering the intensity and local contrast of TIR ship targets, the IFSM, and BCSM are fused to differentiate potential ship targets and suppress heavy sea clutter. Additionally, considering the contour and shape features of TIR ship targets, the STAEM and the statistical shape knowledge constraint are introduced to characterize true ship targets and eliminate residual non-ship targets. Besides, a dual approach is adopted to simultaneously detect both bright and dark ship targets in TIR image.

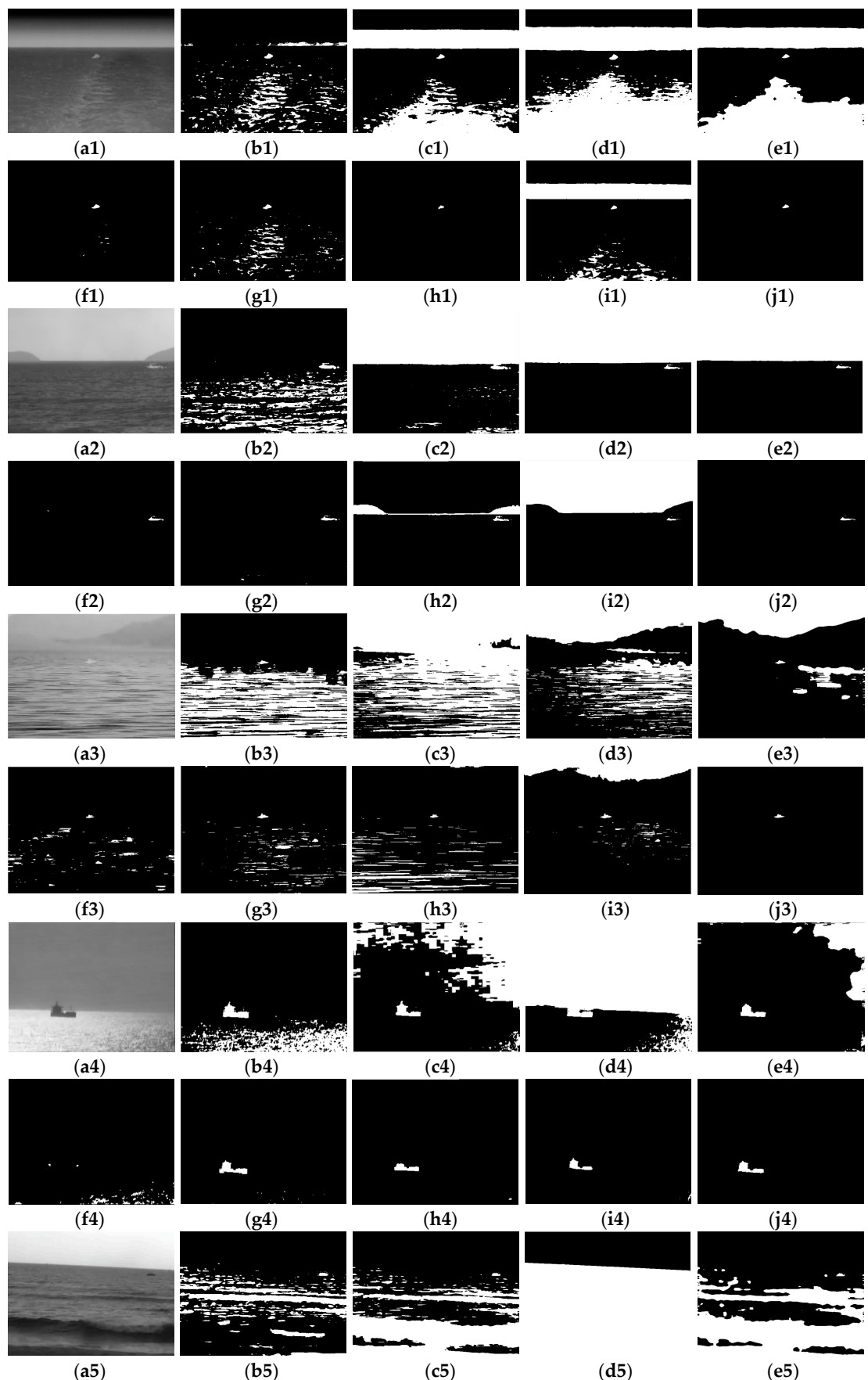

**Figure 14.** *Cont.*

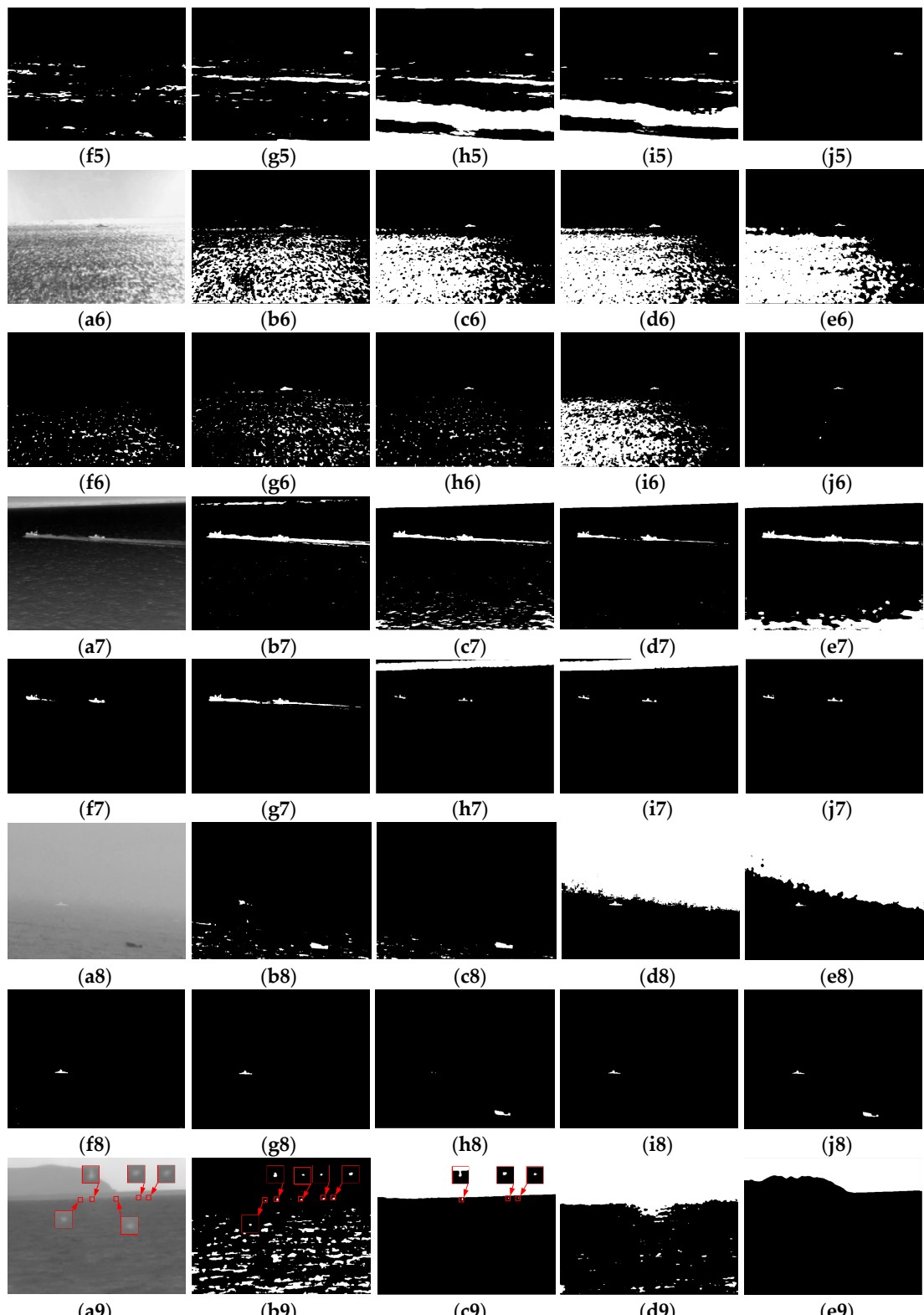

**Figure 14.** *Cont*.

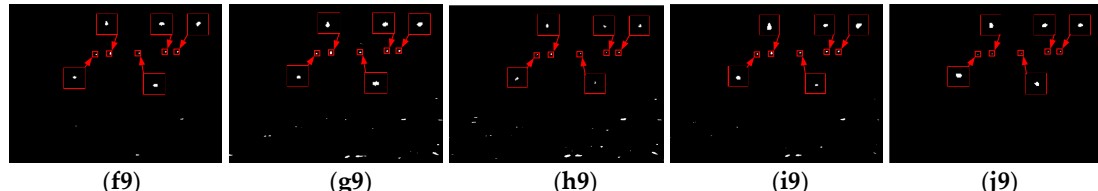

| (f9) | (g9) | (h9) | (i9) | (j9) |

**Figure 14.** The detection results of different methods for the Figure 12. (**a1–a9**) show the representative frames of the nine TIR sequences, respectively, (**b1–b9**) are the detection results of THF/BHF, (**c1–c9**) are the detection results of 2DME, (**d1–d9**) are the detection results of 2DO, (**e1–e9**) are the detection results of MSS, (**f1–f9**) are the detection results of IMFS, (**g1–g9**) are the detection results of IFCM, (**h1–h9**) are the detection results of CVM, (**i1–i9**) are the detection results of HCST, (**j1–j9**) are the detection results of proposed method.

4.2.3. Quantitative Comparison to TIR Ship Target Detection Baseline Methods

To evaluate the performance of TIR ship target detection methods in the experiments, misclassification error (ME) [3], relative foreground area error (RAE) [3], missing alarm ratio (MAR) [4] and false alarm ratio (FAR) [4] are utilized as quantitative evaluation metrics. ME denotes the percentage of pixels that are wrongly classified, that is, the background misclassified as foreground and the foreground misclassified as background. RAE represents the detected area accuracy between the detected image and the ground-truth image. MAR reflects the probability of missed targets among which ship targets truly exist and FAR illustrates the rate of detected targets where ship targets do not exist. Hence, smaller ME, RAE, MAR, and FAR imply the better results, and defined as:

$$ME = 1 - \frac{|B_O \cap B_T| + |F_O \cap F_T|}{|B_O| + |F_O|} \tag{28}$$

$$RAE = \begin{cases} \frac{A_O - A_T}{A_O}, & A_T < A_O \\ \frac{A_T - A_O}{A_T}, & A_T \geq A_O \end{cases}, \tag{29}$$

$$MAR = \frac{MT}{MT + DT}, \tag{30}$$

$$FAR = \frac{FD}{DT + FD}, \tag{31}$$

where $B_O$ and $F_O$ are the background pixels and ship target pixels in the ground-truth image (manually labeled), respectively. $B_T$ and $F_T$ are the background pixels and ship target pixels of detected image, respectively. $|\cdot|$ represents cardinality of a set. $A_O$ is the area of true target (manually labeled), and $A_T$ is the area of detected ship target. $MT$ is the number of missed targets, $DT$ is the number of detected targets, and $FD$ is the number of false detection results. The average values of ME, RAE, MAR, and FAR of different methods for various sequences are listed in Table 4. Table 4 shows that the proposed method obtains smaller average values for ME, RAE, MAR, and FAR on all nine sequences than the other compared eight methods. Consequently, the experimental results verify that the proposed method not only has better ship target detection capability compared with other classical and state-of-the-art methods, but also can work stably for different complex maritime scenarios.

**Table 4.** The detection performance of different methods for various sequences with 4500 TIR ship images.

| Metrics | Methods | Seq1 | Seq2 | Seq3 | Seq4 | Seq5 | Seq6 | Seq7 | Seq8 | Seq9 | Average |
|---------|---------|------|------|------|------|------|------|------|------|------|---------|
| ME | THF/BHF | 0.0826 | 0.1341 | 0.3606 | 0.0545 | 0.2172 | 0.2433 | 0.0383 | 0.0126 | 0.0882 | 0.1368 |
| | 2DME | 0.3133 | 0.4516 | 0.6720 | 0.2610 | 0.5216 | 0.2858 | 0.0815 | 0.0097 | 0.1849 | 0.3090 |
| | 2DO | 0.4017 | 0.4194 | 0.3531 | 0.7029 | 0.7139 | 0.4183 | 0.0710 | 0.6084 | 0.2778 | 0.4729 |
| | MSS | 0.4589 | 0.4214 | 0.1508 | 0.0517 | 0.3813 | 0.3370 | 0.1094 | 0.5073 | 0.1365 | 0.2838 |
| | IMFS | 0.0048 | 0.0032 | 0.0191 | 0.0170 | 0.0243 | 0.0289 | 0.0030 | 0.0018 | 0.0009 | 0.0114 |
| | IFCM | 0.0405 | 0.0047 | 0.0522 | 0.0091 | 0.0462 | 0.0899 | 0.0217 | 0.0022 | 0.0014 | 0.0298 |
| | CVM | 0.0018 | 0.0257 | 0.0954 | 0.0105 | 0.2143 | 0.0523 | 0.0597 | 0.0016 | 0.0013 | 0.0514 |
| | HCST | 0.1285 | 0.3937 | 0.1069 | 0.0062 | 0.1635 | 0.2187 | 0.0315 | 0.0025 | 0.0011 | 0.1170 |
| | **Proposed** | **0.0016** | **0.0023** | **0.0031** | **0.0045** | **0.0104** | **0.0047** | **0.0018** | **0.0012** | **0.0005** | **0.0033** |
| RAE | THF/BHF | 0.3978 | 0.6005 | 0.3553 | 0.3297 | 0.3099 | 0.4198 | 0.9341 | 0.3456 | 0.7458 | 0.4932 |
| | 2DME | 0.1705 | 0.8735 | 0.9774 | 0.2953 | 0.1065 | 0.0064 | 0.8817 | 0.2447 | 0.9999 | 0.5062 |
| | 2DO | 0.2294 | 0.3887 | 0.8650 | 0.9629 | 0.9869 | 0.5350 | 0.8243 | 0.7093 | 1.0000 | 0.7224 |
| | MSS | **0.1152** | 0.1918 | 0.1198 | 0.1551 | 0.0856 | 0.1168 | 0.9049 | 0.7077 | 1.0000 | 0.3774 |
| | IMFS | 0.1731 | 0.1817 | 0.1905 | 0.9930 | 0.9962 | 1.0000 | 0.5285 | 0.6018 | 0.3077 | 0.5525 |
| | IFCM | 0.2196 | 0.2784 | 0.4267 | 0.2558 | **0.0208** | 0.0525 | 0.8772 | 0.5963 | 0.4223 | 0.3500 |
| | CVM | 0.2607 | 0.2865 | 0.1570 | 0.3008 | 0.1106 | 0.2683 | 0.2836 | 0.1527 | 0.5982 | 0.2687 |
| | HCST | 0.1667 | 0.2809 | 0.1878 | 0.1103 | 0.3830 | 0.3252 | 0.1175 | 0.6994 | **0.0217** | 0.2547 |
| | **Proposed** | 0.1203 | **0.1523** | **0.0992** | **0.0874** | 0.0213 | **0.0040** | **0.0204** | **0.0092** | 0.1112 | **0.0695** |
| MAR | THF/BHF | 0.5220 | **0.0000** | 0.4620 | 0.3540 | 0.4480 | 0.5440 | 0.8900 | 0.5070 | **0.0000** | 0.4141 |
| | 2DME | 0.2160 | 0.7560 | 1.0000 | 0.0460 | 0.4120 | 0.4320 | 0.6850 | 0.5320 | 0.6412 | 0.5245 |
| | 2DO | 0.4480 | 0.4740 | 0.7260 | 0.6640 | 1.0000 | 0.5120 | 0.6033 | 0.7870 | 1.0000 | 0.6905 |
| | MSS | 0.2020 | 0.3200 | 0.2480 | 0.0780 | 0.3340 | 0.3980 | 0.8233 | 0.6630 | 1.0000 | 0.4518 |
| | IMFS | **0.0000** | **0.0000** | **0.0640** | 1.0000 | 1.0000 | 1.0000 | **0.0000** | 0.5000 | 0.0200 | 0.3982 |
| | IFCM | 0.0014 | **0.0000** | 0.1040 | 0.0400 | 0.1060 | 0.1560 | 0.3216 | 0.5000 | 0.0014 | 0.1367 |
| | CVM | **0.0000** | 0.3480 | 0.1920 | 0.1240 | 0.2620 | 0.2800 | **0.0000** | 0.4190 | 0.0796 | 0.1894 |
| | HCST | 0.0420 | 0.0220 | 0.0780 | 0.1020 | 0.1460 | 0.1880 | **0.0000** | 0.5020 | **0.0000** | 0.1200 |
| | **Proposed** | **0.0000** | **0.0000** | 0.0760 | **0.0280** | **0.0720** | **0.0840** | **0.0000** | **0.0150** | 0.0010 | **0.0307** |
| FAR | THF/BHF | 0.9688 | 0.8485 | 0.9661 | 0.9160 | 0.9574 | 0.9728 | 0.8925 | 0.9490 | 0.7959 | 0.9186 |
| | 2DME | 0.8734 | 0.8913 | 1.0000 | 0.8077 | 0.8987 | 0.9123 | 0.8758 | 0.7895 | 0.7756 | 0.8694 |
| | 2DO | 0.8667 | 0.7500 | 0.9627 | 0.7143 | 1.0000 | 0.9225 | 0.6667 | 0.7917 | 1.0000 | 0.8527 |
| | MSS | 0.5455 | 0.6667 | 0.8285 | 0.7187 | 0.6875 | 0.8701 | 0.6364 | 0.8091 | 1.0000 | 0.7514 |
| | IMFS | 0.3750 | 0.1255 | 0.8433 | 1.0000 | 1.0000 | 1.0000 | 0.2568 | 0.1395 | 0.3406 | 0.5645 |
| | IFCM | 0.8990 | 0.3108 | 0.9190 | 0.5326 | 0.8724 | 0.9563 | 0.3529 | 0.4924 | 0.6058 | 0.6601 |
| | CVM | **0.0556** | 0.4167 | 0.9074 | 0.4647 | 0.8362 | 0.8964 | 0.3726 | 0.1512 | 0.6551 | 0.5284 |
| | HCST | 0.8592 | 0.4527 | 0.8571 | 0.2581 | 0.7742 | 0.9314 | 0.2863 | 0.1427 | 0.5739 | 0.5706 |
| | **Proposed** | 0.0579 | **0.0338** | **0.2007** | **0.0000** | **0.1853** | **0.6717** | **0.1959** | **0.0285** | **0.1638** | **0.1708** |

## 5. Conclusions and Future Work

A new method based on gray-level morphological reconstruction and multi-feature analysis is proposed in this paper to detect small ship targets under heavy maritime background clutter. The proposed TIR ship target detection method can automatically segment the small ship target from the sea background clutter which has intricate texture and strong contrast. By using the opening or closing based GMR, the intricate sea clutter is removed while the intensity, shape, and contour information of ship target are retained, so the proposed ship target detection method is robust to heavy sea clutter. Considering the intensity and contrast features of TIR ship targets after GMR-based pretreatment, the IFSM and BCSM are computed and fused, so the potential ship targets can be well highlighted and the complex clutter can be suppressed simultaneously. Furthermore, considering the contour and shape features of TIR ship targets, a two-step ship verification strategy including STAEM-based contour descriptor and the statistical shape knowledge constraint is constructed, so the true ship targets are efficiently extracted from residual non-ship clutter. Moreover, the dual approach is applied by directly adding bright and dark ship target maps, so both bright and dark ship targets in TIR image can be simultaneously detected. Extensive experiments verify that the proposed small ship detection algorithm has a better detection performance than compared state-of-the-art methods, including THF/BHF, 2DME, 2DO, MSS, IMFS, IFCM, CVM, and HCST. The experimental results also

demonstrate that the proposed method can work stably for ship target with unknown brightness, variable quantities, sizes, and shapes.

However, although the proposed method has achieved considerable detection results, the method is based on the assumption that ship targets are viewed as uniform regions under the sea background in thermal infrared images due to long imaging distance, so it cannot work well for segmenting the whole ship target with uneven intensities in near-distance or near-infrared imaging. Therefore, combining the proposed method with region growing method [47] to develop a high-quality ship segmentation algorithm for ship targets with uneven intensities is one important direction in our future studies. Besides, the proposed method can accurately detect both bright and dark ship targets by directly adding bright and dark ship target maps in most cases but may cause some false alarms in the ship target detection submerged in dense sun-glint clutter, as Figure 14j6 illustrates. Therefore, combining multi-frame verification [48] or deep multi-feature fusion [27,49] strategy to recognize ship target submerged in heavy sun-glint clutter is our another important research direction.

**Author Contributions:** Y.L. and Z.L. proposed the ship target detection method. Y.L. conducted the experiments and wrote the manuscript under the supervision of Z.L. and Y.H. Y.Z., B.L. and W.X. assisted in the database selection and performed the comparison experiments. All the authors have read and revised the final manuscript.

**Funding:** This work was partially supported by the National Natural Science Foundation of China under Grant No. 61675036, Chinese Academy of Sciences Key Laboratory of Beam Control Fund under Grant No. 2017LBC006, Fundamental Research Funds for Central Universities under Grant No. 2018CDGFTX0016, Chongqing Research Program of Basic Research and Frontier Technology under Grant No. CSTC2016JCYJA0193.

**Acknowledgments:** We would like to express our thanks to anonymous reviewers for their insightful and valuable suggestions, which helped us to improve the quality of this paper.

**Conflicts of Interest:** The authors declare no conflict of interest.

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
