# Peer review of "Thermal Infrared Small Ship Detection in Sea Clutter Based on Morphological Reconstruction and Multi-Feature Analysis"

_applsci, doi:10.3390/app9183786_

Round 1
Reviewer 1 Report
Overall the paper provides enough contribution to be of impactful read to the target audience.
I have two comments if these could be incorporated.
A block diagram early in the paper regarding the method would greatly help in providing the overview of the remaining details. The English writing of the paper could be improved to convey the message across not only more clearly but also interesting for the reader.
Reviewer 2 Report
Author have proposed an algorithm for ship detection for clutter scenes from infrared data. They have shown their method performs better than few other methods on only one dataset.
I have following quetries.
- Why author claims their method is suitable for infrared only? What differentiate this method to be exclusive for infrared and not for visible and infra red video/images?
- The author claims their method can work for clutter but the data they have shown does not have much maritime traffic clutter. A better state-of-the-art dataset will be Singapore maritime dataset. This dataset has sufficient clutter and videos for the similar scene both for visible and infra red.
- The author's claim in the abstract that proposed method can work for unknown brightness, unknown number of targets and unknown sizes and shapes of ship are not verified with sufficient quantitative proof.
- Following reference have more work for literature review and discussions.
a. Kristan, M., Perš, J., Sulič, V., & Kovačič, S. (2014, November). A graphical model for rapid obstacle image-map estimation from unmanned surface vehicles. In Asian Conference on Computer Vision (pp. 391-406). Springer, Cham.
b. Prasad, D. K., Rajan, D., Rachmawati, L., Rajabally, E., & Quek, C. (2017). Video processing from electro-optical sensors for object detection and tracking in a maritime environment: a survey. IEEE Transactions on Intelligent Transportation Systems, 18(8), 1993-2016.
c. Ribeiro, Ricardo, Gonçalo Cruz, Jorge Matos, and Alexandre Bernardino. "A Dataset for Airborne Maritime Surveillance Environments." IEEE Transactions on Circuits and Systems for Video Technology (2017).
d. Prasad, D. K., Prasath, C. K., Rajan, D., Rachmawati, L., Rajabally, E., & Quek, C. (2018). Object detection in a maritime environment: Performance evaluation of background subtraction methods. IEEE Transactions on Intelligent Transportation Systems, 20(5), 1787-1802.
- Why author choose to classical feature based method for ship detection and didn't tried simple deep learning architecture like CNN,Alexnet,YOLO,ResNet,Inception??
e. Chen, W., Li, J., Xing, J., Yang, Q., & Zhou, Q. (2018, August). A maritime targets detection method based on hierarchical and multi-scale deep convolutional neural network. In Tenth International Conference on Digital Image Processing (ICDIP 2018) (Vol. 10806, p. 1080616). International Society for Optics and Photonics.
f. Chen, Y., Chen, X., Zhu, J., Lin, F., & Chen, B. M. (2018, October). Development of an Autonomous Unmanned Surface Vehicle with Object Detection Using Deep Learning. In IECON 2018-44th Annual Conference of the IEEE Industrial Electronics Society (pp. 5636-5641). IEEE.
g. Moosbauer, S., Konig, D., Jakel, J., & Teutsch, M. (2019). A Benchmark for Deep Learning Based Object Detection in Maritime Environments. In Proceedings of the IEEE Conference on Computer Vision and Pattern Recognition Workshops (pp. 0-0).
VAIS dataset is small set other dataset has more than 200k target images under various illumination and environmental condition. The robustness of the proposed method can be better verified under larger dataset like Singapore maritime dataset or others.
Round 2
Reviewer 2 Report
The author revised version is reasonable but still they have not yet addressed my first query concretely.
Why author claims their method is suitable for infrared only? What differentiate this method to be exclusive for infrared and not for visible and infra red video/images?
Author response is that LWIR image will have good signature of small ships. Yes I agree to it. But they are yet not able to answer that why they claim their method is for infra-red and can not be applied to visible and infra red data.
My view point is that the proposed ,method is generic one and can be applied to any type of images. The proposed method is not based on specific mathematics of Electromagnetics of LWIR so title claiming infra red is not justified.
It might be that their proposed algorithm is more tailored to LWIR dataset of VAIS and performance on other infra red dataset will be poor. But still it will be good to see the performance on other dataset as well.
It will be great if author would have given quantitative proof that the proposed method is only designed for LWIR and fundamentally its incorrect to apply the proposed method to any other types of data(NIR,MWIR,Visible) . It will be better to see the evaluation on other dataset as well.
- The VAIS dataset still do not have clutter like other dataset so title claiming clutter has also not been verified.
